# VideoGUI: A Benchmark for GUI Automation from Instructional Videos

**Kevin Qinghong Lin**[1], **Linjie Li**[2], **Difei Gao**[1], **Qinchen Wu**[1],
**Mingyi Yan**[1], **Zhengyuan Yang**[2], **Lijuan Wang**[2], **Mike Zheng Shou**[1✉]
[1]Show Lab, National University of Singapore    [2]Microsoft

## Abstract

Graphical User Interface (GUI) automation holds significant promise for enhancing human productivity by assisting with computer tasks. Existing task formulations primarily focus on simple tasks that can be specified by a single, language-only instruction, such as "Insert a new slide." In this work, we introduce **VideoGUI**, a novel multi-modal benchmark designed to evaluate GUI assistants on visual-centric GUI tasks. Sourced from high-quality web instructional videos, our benchmark focuses on tasks involving professional and novel software (*e.g.,* Adobe Photoshop or Stable Diffusion WebUI) and complex activities (*e.g.,* video editing). VideoGUI evaluates GUI assistants through a *hierarchical* process, allowing for identification of the specific levels at which they may fail: (*i*) **high-level planning:** reconstruct procedural subtasks from visual conditions without language descriptions; (*ii*) **middle-level planning:** generate sequences of precise action narrations based on visual state (*i.e.,* screenshot) and goals; (*iii*) **atomic action execution:** perform specific actions such as accurately clicking designated elements. For each level, we design evaluation metrics across individual dimensions to provide clear signals, such as individual performance in clicking, dragging, typing, and scrolling for atomic action execution. Our evaluation on VideoGUI reveals that even the SoTA large multimodal model GPT4o performs poorly on visual-centric GUI tasks, especially for high-level planning. The data and code are available at https://github.com/showlab/videogui.

## 1 Introduction

In the digital age, individuals rely on computers for a vast array of daily activities (*e.g.,*web browsing, entertainment etc.). These activities often necessitate the use of diverse software, which are accessed primarily through Graphical User Interfaces (GUIs). Large language models (LLMs) [1], which excel in understanding complex language instructions and integrating various tools seamlessly, have shown great potential in GUI automation [2, 3, 4, 5]. They could streamline the navigation of digital interfaces and significantly enhance productivity, *e.g.,*assisting slide template creation in Powerpoint with just a few keywords [2].

Recently, notable efforts have been made in GUI automation evaluation, benchmarking model performances on Web [4, 6, 7] or Smartphone GUI navigation [8, 9], given screenshots or HTML codes [10, 11]. Follow-up works [12, 13, 14] develop executable environments with well-defined action spaces, which removes the dependencies on pre-defined inputs. Nonetheless, most existing GUI benchmarks [15, 16] restrict their applications to simpler domains and tasks that can be described with a single text instruction (*e.g.,* "Insert a new slide on the second page"). In real-world scenarios, users rarely struggle with basic operations that can be clearly described in text. Rather, they often encounter difficulties in performing novel and advanced tasks (*e.g.,* "Create a special animation effects in powerpoint"), which extend far beyond basic operations, and rely more on visual signals than text instructions to complete such tasks.

---

✉: Corresponding Author.

38th Conference on Neural Information Processing Systems (NeurIPS 2024) Track on Datasets and Benchmarks.

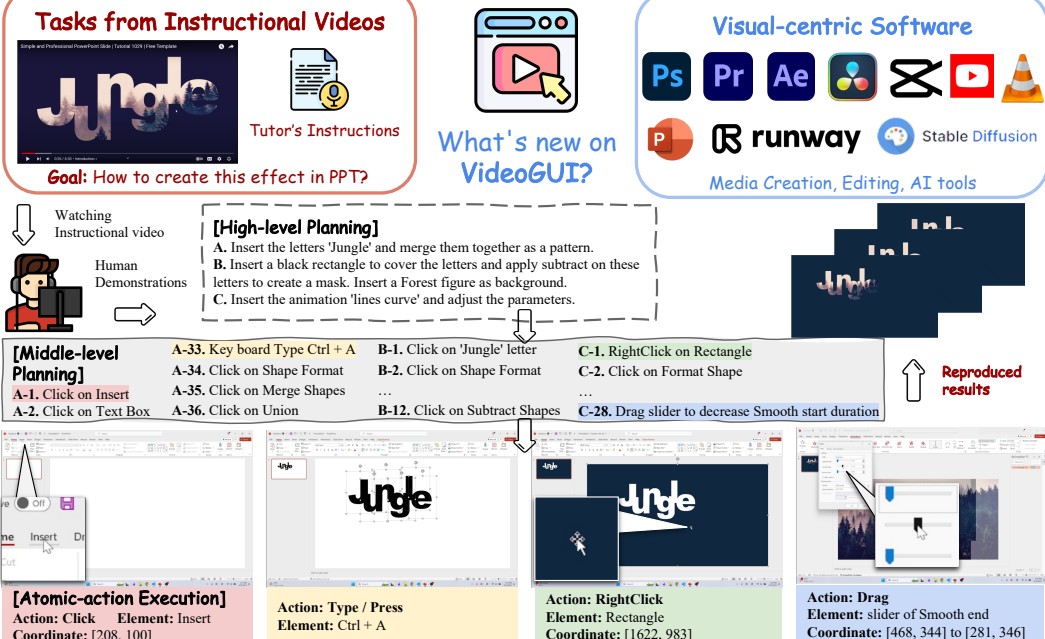

**Figure 1: A brief illustration of VideoGUI.** VideoGUI focuses on professional and novel software like PR, AE for video editing, and Stable Diffusion, Runway for visual creation. We source tasks from high-quality instructional videos (see an example here), with annotators replicating these to reproduce effects; We provide detailed annotations with planning procedures and recorded actions for hierarchical evaluation.

Inspired by the abundant instructional videos that teach average users for performing novel and complex GUI tasks, we introduce VideoGUI, a new multi-modal GUI benchmark derived from high-quality web instructional videos. As shown in Fig. 1, VideoGUI provides high- quality annotations by having participants reproducing the instructional videos, capturing multi-level labels from procedural planning to atomic actions with element locations. VideoGUI covers 11 visual-centric software applications and features 86 complex tasks (averaging 22.7 actions each) and 463 subtasks, alongside hierarchical manual planning and 2.7K manual action annotations (Tab. 1).

With VideoGUI, we propose a comprehensive evaluation suite for GUI assistants via a *hierarchical* process: ($i$) high-level planning involves reconstructing procedural subtasks from visual cues without language descriptions; ($ii$) middle-level planning details the steps for completing a subtask with a sequence of precise action narrations based on visual state and textual query; ($iii$) atomic action execution is to perform the target actions (*e.g.,*click on the designated element). For each level, we design evaluation metrics across *individual dimensions* to assess model performance, which help to pinpoint model limitations.

We conduct comprehensive evaluation of SoTA large multimodal models (LMMs) on VideoGUI, and find that even the current best model GPT-4o fails to complete a single full task in our benchmark. Our empirical results show that the bottleneck surprisingly lies in planning rather than action execution, even though GPT-4o is not known for grounding. Moreover, planning from textual queries is much easier than planning from visual previews for almost all models evaluated, which further implies the difficulty of visual-centric GUI tasks. Our findings shed lights on the directions for developing the next generation of models or agent systems towards GUI automation.

## 2 Related Works

**Benchmarks GUI Tasks.** In recent years, a range of works have focused on modeling GUI tasks and benchmarking agents, which include: ($i$) Web browsing [15], where agents are developed to interact with web interfaces for navigation and to support a variety of tasks like online shopping. ($ii$) Mobile navigation [8], aimed at improving accessibility within mobile GUI simulator environments, such as Android and iOS [21]. ($iii$) Several efforts aimed at resolving issues with computer desktop software

| Benchmark | # Task | Platform | Source | Query format | | | # Avg. Action | Eval. dimension | | |
|---|---|---|---|---|---|---|---|---|---|---|
| | | | | Text | Image | Video | | Task SR. | Hier. Plan. | Action Exec. |
| Mind2Web [6] | 2350 | Web | Screenshot | ✓ | | | 7.3 | ✓ | | ✓ |
| PixelHelp [17] | 187 | Android | Emulator | ✓ | | | 4.2 | ✓ | | ✓ |
| AITW [10] | 30K | Android | Emulator | ✓ | | | 6.5 | ✓ | | ✓ |
| AssistGUI [18] | 100 | Windows | Web Video | ✓ | | | – | ✓ | | |
| OSWorld [19] | 369 | Win.+Ubuntu | Emulator | ✓ | | | – | ✓ | | |
| V-WebArena [20] | 910 | Web | Screenshot | ✓ | ✓ | | – | ✓ | | |
| **VideoGUI** SUBTASK 463 FULLTASK 86 | | Win. +Web | **Video + Human Demonstration** | ✓ | ✓ | ✓ | 5.6 **22.7** | ✓ | ✓ | ✓ |

Table 1: **Comparison of VideoGUI with existing GUI datasets.** VideoGUI differs from existing benchmarks in: (*i*) sourcing from instructional videos with human demonstrations; (*ii*) featuring 86 challenging full tasks averaging 22.7 actions, and 463 subtasks; (*iii*) offering comprehensive evaluation with hierarchical planning and action categories.

have emerged, such as grounding UI elements in offline settings like screenshots [16]. Additionally, there has been development of executable simulated environments [22] for more interactive evaluation. AssistGUI [18] is one project that utilizes video subtitles and metadata from instructional videos as reference, and evaluates the model by determining outcomes based on task success or failure.

Differing from these works, we focus on more complex and challenging GUI tasks that often require individuals to follow instructional videos to replicate long procedure operations and achieve goals. Specifically, We've developed a comprehensive evaluation framework that covers high-level task procedures, mid-level action decomposition, and atomic-level action execution. Our approach emphasizes UI visual-centric perception over textual understanding, focusing on identifying visual goals and transitions between states, which present significant challenges.

**Multi-Modal Agents.** Recent studies have highlighted the promising potential of LLMs beyond language modeling. Notable advancements in Chain of Thought (CoT) [23] and ReAct [24] strategies have demonstrated LLMs' capabilities as autonomous agents, capable of completing complex tasks through dynamic programming [25, 26]. Motivated by these progresses, follow-up works connect LLMs with visual experts to enable multimodal applications, such as visual question answering [27], or image editing applications [28]. In the realm of Embodied GUI tasks, the primary challenges involve understanding complex UI elements and planning to execute diverse tasks. This has led to the development of approaches such as: (*i*) Training-free agent systems, which primarily consist of two stages: the first involves semantically understanding UI elements [29, 30, 31], either by transforming the GUI into HTML representations or language descriptions [11, 32], or using off-the-shelf visual models like OCR [33] and SoM [32, 34]. The second stage involves utilizing LLMs to integrate information and generate responses. This method heavily relies on closed-source LLMs [1], incurring significant costs. Additionally, it limits the model's UI visual perception abilities, such as demonstrating goals or state transitions visually rather than linguistically. (*ii*) Vision-Language-Action models [35, 36], which are pretrained on large-scale GUI vision-text corpus (*e.g.,*screenshots). This enables the LLMs to obtain more abilities such as element grounding and reasoning in unified responses. However, it remains unclear when and how to employ different types of GUI agents or tools. VideoGUI provides a comprehensive suite for studying and benchmarking these models.

## 3 VideoGUI Benchmarks

### 3.1 Data Construction

**Data source.** VideoGUI consists of 11 software applications, categorized into: (*i*) media creation, featuring visual and animation tools like PowerPoint, Runway, and Stable Diffusion; (*ii*) media editing, including Adobe Photoshop, Premiere Pro, After Effects, CapCut, and DaVinci Resolve; (*iii*) media browsing, with platforms like YouTube, VLC Player, and Web Stock.

**Pipeline.** The VideoGUI creation pipeline is illustrated in Fig.2. For each software, **(i)** we manually select instructional videos paired with high-quality transcripts from YouTube, focusing on those teaching practical and novel usages. To collect the *human manipulation trajectory*, we build a simulated environment to monitor user behaviors including `Click`, `Drag`, `Type/Press`, and `Scroll`. **(ii)** We invite five participants who first watch the selected video and then try to reproduce the effects shown using our simulator, which records all cursor and keyboard activities (*e.g.,* $[x, y]$ coordinates

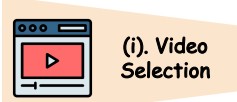 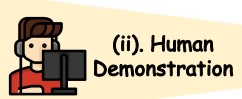 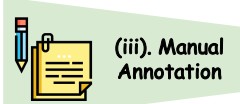 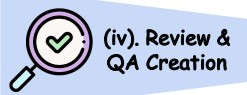

**Figure 2: Illustration of VideoGUI creation pipeline**, encompassing four phases: **(i)** High-quality instruction videos are manually selected, **(ii)** Participants replicate skills demonstrated in videos, **(iii)** Participants annotate task elements and procedures, **(iv)** Annotated data is validated manually for VideoGUI benchmarking use.

of a `RightClick`). Afterward, they provide a brief description of the overall goal for the full task, which can be optionally used as text query during evaluation. Then the operations shown in the video is broken down into several *subtasks* and annotated with textual descriptions, each focusing on a main functionality operation (*e.g.,* inserting a figure). **(iii)** We also instruct the annotators to identify the active elements (*e.g.,* buttons 'Insert') for each action, as they are not automatically identified and recorded by our simulator. After the demonstration, we retain all available files, including material, project files, and visual outcomes (the latter being our *full-task's visual query*). **(iv)** The participants cross-validate the annotations, and remove unclear/incorrect ones.

**Data statistic.** Overall, VideoGUI includes 178 tasks across 11 software applications (Fig. 3a) on Windows and Web browsers (Chrome, Edge, Firefox). It comprises 86 complex tasks (*i.e.,* full task) and 92 simple tasks (*i.e.,* subtask) that do not require high-level planning, where those 86 full tasks can be further divided into 371 subtasks, resulting in a total of 463 subtasks. Fig. 3b shows the distribution of number of actions per task. In total, we collect 2,712 atomic manual actions. As shown in Fig. 3c, the most common action is LeftClick (66.2%), while RightClick and Scroll are the least common actions (approximately 2%).

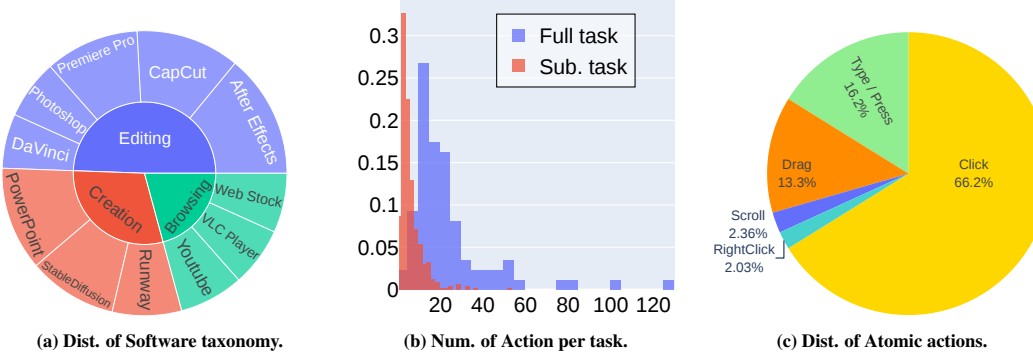

(a) Dist. of Software taxonomy.    (b) Num. of Action per task.    (c) Dist. of Atomic actions.

**Figure 3: Data statistics of VideoGUI.**

## 3.2 Evaluation and Metrics

**Overview.** Imagine a human to complete the complex task illustrated in Tab. 2, we often first break down the full task into sub-tasks, and then sequentially perform the actions required to complete each subtask. Existing GUI benchmarks [6, 17, 18] predominantly use a boolean metric (i.e., Success Rate) to measure the success of completing a task. It may work okay for simpler tasks involving only a few actions, but is clearly not sufficient in providing feedback on where the models fall short, especially as the complexity of the task increases (*e.g.*, a full task with over 100 actions), and nonetheless to say to guide future improvements in modeling for GUI navigation.

To address this, we propose a *hierarchical* decomposition of tasks into three key stages: A. High-level Planning, which translates task instructions or reverse engineers final outcomes into several key milestones. B. Middle-level Planning, which converts each milestone into detailed action narrations. C. Atomic-level Execution, which focuses on accurately executing specific actions, such as clicking and typing, as dictated by the narration. The whole evaluation scheme is shown in Fig. 4. Each part is discussed in detail subsequently.

⬤ **High-level Planning**. This method translates instructions or outcomes into key milestones (*i.e.*, subtasks). Unlike previous approaches that start with explicit textual queries, practical scenarios often rely on final visual demonstrations like animations, requiring the reconstruction of past procedural tasks. Accordingly, we develop three categories based on different modal inputs:

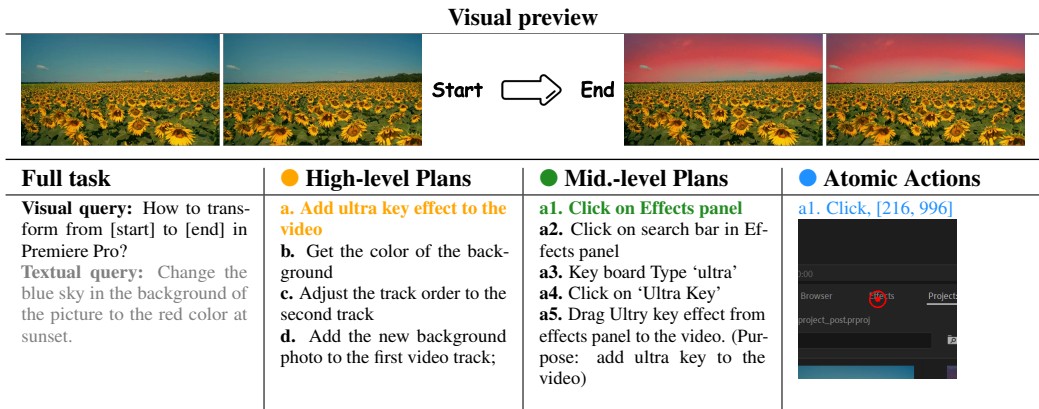

**Table 2:** *Hierarchical* annotations in VideoGUI (Premiere Pro). The top row displays the video input and the desired task outcome as the visual query, with an optional textual query describing the video editing effect. The model is expected to "reverse-engineer" this outcome through a hierarchical process: first by planning high-level milestones (i.e., sub-tasks), then detailing each milestone into step-by-step narrations at the middle level, and finally translating these narrations into executable actions.

- *Visual query* is our primary setting, with only a visual preview are provided, for example, two photos before and after editing with Photoshopor an animation effect created in PowerPoint.
- *Textual query* explicitly defines the objectives using detailed descriptions.
- *Visual query + Textual query*, which provides the most complete information.

**Metrics**: Planning involves open-ended question-answering with multiple correct approaches, making traditional metrics insufficient. To adaptively evaluate model responses, we define a critic using GPT-4-Turbo [37] inspired by [38], prompting LLM to focus on key elements and operations such as 3D shape and specific animation types. We score the model's generated procedure steps against the ground truth on a scale from $0$ (totally unrelated) to $5$ (almost correct).

🟢 **Middle-level Planning**. Given a milestone task, the agent should perform appropriate UI operations based on its observation state (e.g., screenshots). This stage aims to generate a sequence of precise action narrations (*i.e.,* desired action type with an accurate element) by combining textual milestones and visual observations. We devise three modes:

- *Visual initial state + Textual query:* Our main setting, as it accepts the output from the previous high-level planning, and the initial state (*i.e.,* screenshot) can be straightforwardly obtained.
- *Textual query:* A common setting in most existing works.
- *Visual state transition (initial and end state):* the most challenging setting requiring the model to understand differences by screenshot transition and reverse its fine-grained actions.

**Metrics**: similar to the high-level planning phrase, we use the LLM as the critic for scoring.

🔵 **Atomic-action Execution**. After planning, the agent should respond to the action narrations with middle-level planning. We evaluate whether the model can accurately interpret narrations and perform the corresponding actions. Specifically, we consider the four most common action categories:

- `Click`: For a narration like `Click on the Insert button`, the model must accurately localize desired element on the screenshot by providing a bounding box or click position $[x, y]$.

  **Metrics:** ($i$) Dist $:= \frac{\Delta}{L}$, where $\Delta$ is the pixel difference between the predicted location and the ground-truth coordinate. $L$ is the farthest distance from the ground-truth location to any of the four screenshot vertices for adaptive normalization. ($ii$) Recall@$d$: In practice, a click is usually valid when it falls within a very short distance, such as within a button area. We calculate the recall score with a threshold $d$, which we empirically set to 100.

- `Drag`: can be approximated as a combination of a Click with a short-phrase movement, where the model must infer both the initial and end coordinates.

  **Metrics**: ($i$) Dist $:= \frac{1}{2}\left(\frac{\Delta s}{Ls} + \frac{\Delta e}{Le}\right)$ by jointly measuring the start and end distance, then averaging them. ($ii$) Recall@$d$: This metric is more stricter than click, requiring both the start and end predictions to be within a distance $d$; otherwise, the score is 0.

- `Scroll`: The scroll action assesses whether the desired elements are visible in the current screenshot, determining if a scroll action is needed (e.g., scroll up, scroll down, or no need).

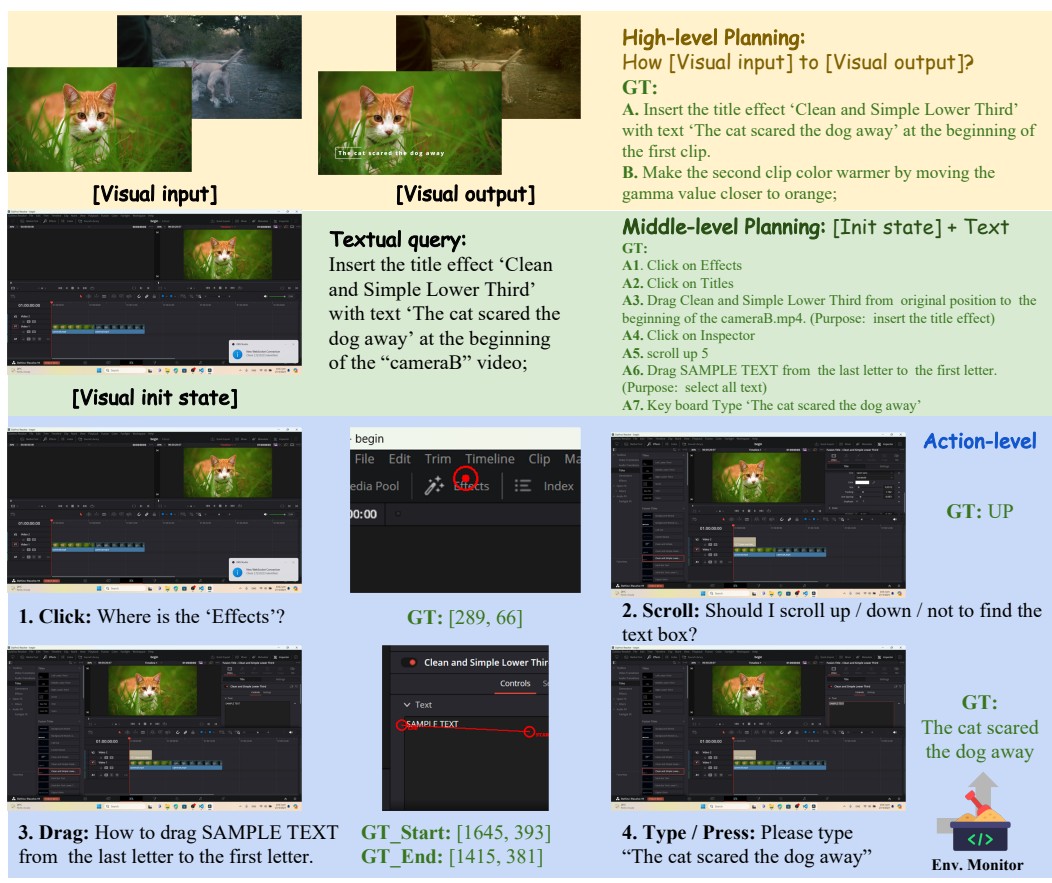

Figure 4: **Illustration of VideoGUI hierarchical evaluation,** containing high-level planning (visual query), mid-level planning (visual state + text query), and action-level execution (with 4 types: click, scroll, drag, type/press). We use DaVinci software as an example.

**Metrics**: We frame this QA as a multiple-choice question: 'Could you advise whether I need to scroll to see [target element]?' with options: [A] No need to scroll; [B] Scroll up; [C] Scroll down. To prevent bias, we shuffle the choices randomly and calculate the accuracy score.

- `Type & Press`: For type actions such as `Type 'Hello world!'`, the agent must accurately produce the string through keystrokes. For commands like `Ctrl+C`, it must execute multiple keystrokes and button presses. Most GUI agents utilize PyAutoGUI [39] for these operations, framing them as coding challenges that require verification for correctness.

**Metrics**: We design a *Sandbox* scheme by developing a mini simulator that executes the code produced by the agent. Additionally, we use a monitor to listen for the keys pressed or typed. We then compare the monitored results with the ground-truth results to check for matches. This setting is evaluated using Recall (*i.e.,* whether the GT is produced) and Precision (*i.e.,* the count number of GT and actual outputs to study redundancy).

## 4 Experiments

### 4.1 Baseline settings

We evaluate leading Multi-modal Large Language Models (MLLMs) including GPT-4-Turbo [37], GPT-4o [37], Claude-3-Opus [40], Gemini-Pro-V [41], Qwen-VL-Max [42], and the open-source CogAgent [35]. We also include text-only LLMs such as GPT-3.5-Turbo [43], LLama3-70B [44]and Mixtral-8x22B [45]. Tab. 3 summarizes all evaluated models and their supported modalities.

## 4.2 Main Results on VideoGUI

In Tab. 3, we provide a comprehensive evaluation of baseline models on VideoGUI. Scores are reported for high-level planning (visual query), middle-level planning (visual+text for MLLMs, or text only for LLMs), and atomic action (covering four categories), as well as an overall score summing these three. The planning score was originally evaluated on a scale from 0 to 5. we aimed to provide an overall score that considers both planning and action execution, with the latter being evaluated on a scale of 0 to 1. To achieve this, we normalized the planning scores by dividing by the maximum value of 5. The lowest scores in high-level planning across all models highlight the challenge posed by vision preview instructions. Overall, GPT-4o achieved the highest score of 39.4, excelling in all three tracks. In addition, we incorporate a few simple agent baselines, which use GPT-4T/GPT-4o for high-level/middle-level plan, while incorporate additional tools (*i.e.,*OCR or SoM) to aid its action execution. The use of tools further boosts the overall model performance by $\sim 3$ points for GPT-4o and $\sim 5$ points for GPT-4T. We next dive into the detailed evaluation of procedural planning and action execution for a deeper analysis.

| Model | Support Interleaved Instructions? | | | VideoGUI Evaluation (%) | | | |
|---|---|---|---|---|---|---|---|
| | Text | Image (1f) | Media ($> 1$f) | **High Plan** | **Mid. Plan** | **Action** | **Overall** |
| LLama3-70B [44] | ✓ | | | – | 40.5 | 20.3 | 20.3 |
| Mixtral-8x22B [45] | ✓ | | | – | 36.0 | 19.6 | 18.6 |
| GPT-3.5-Turbo [43] | ✓ | | | – | 49.1 | 22.3 | 23.8 |
| CogAgent [35] | ✓ | ✓ | | 4.4 | 21.8 | 7.4 | 11.2 |
| Qwen-VL-Max [42] | ✓ | ✓ | ✓ | 5.1 | 35.7 | 28.9 | 23.2 |
| Gemini-Pro-V [41] | ✓ | ✓ | ✓ | 7.9 | 28.6 | 23.8 | 20.1 |
| Claude-3-Opus [40] | ✓ | ✓ | ✓ | 9.7 | 45.6 | 39.4 | 31.6 |
| GPT-4-Turbo [37] | ✓ | ✓ | ✓ | 14.3 | 52.9 | 34.4 | 33.9 |
| GPT-4o [37] | ✓ | ✓ | ✓ | **17.1** | **53.5** | **47.6** | **39.4** |
| GPT-4T + OCR | ✓ | ✓ | ✓ | 14.3 | 52.9 | 49.2 | 38.8 |
| GPT-4T + SoM [32] | ✓ | ✓ | ✓ | 14.3 | 52.9 | 44.2 | 37.1 |
| GPT-4o + OCR | ✓ | ✓ | ✓ | 17.1 | 53.5 | **56.3** | **42.3** |
| GPT-4o + SoM [32] | ✓ | ✓ | ✓ | 17.1 | 53.5 | 54.3 | 41.6 |

**Table 3: Full evaluation on VideoGUI with Baselines and their supported *interleaved* instructions**, which might be a text query, an image (1 frame), or a media (more than 1 frame) such as two photos, one or two videos. The bottom block features 4 simple agent baseline, which use GPT-4T/GPT-4o for high-level/middle-level plan, while incorporate additional tools (*i.e.,*OCR or SoM) for action execution.

| Model | High-level Planning $(0-5)$ | | | Middle-level Planning $(0-5)$ | | |
|---|---|---|---|---|---|---|
| | **Vision** | Text | Vision & Text | Vision | Text | **Vision & Text** |
| LLama3-70B [44] | – | 2.62 | – | – | 2.02 | – |
| Mixtral-8x22B [45] | – | 2.43 | – | – | 1.80 | – |
| GPT-3.5-Turbo [43] | – | 2.67 | – | – | 2.46 | – |
| CogAgent [35] | 0.22 | 1.12 | 1.23 | – | 1.32 | 1.09 |
| Qwen-VL-Max [42] | 0.25 | 2.30 | 1.96 | 0.70 | 1.72 | 1.79 |
| Gemini-Pro-Vision [41] | 0.39 | 2.35 | 1.45 | 0.34 | 1.61 | 1.43 |
| Claude-3-Opus [40] | 0.48 | 2.54 | 2.17 | 0.66 | 2.26 | 2.28 |
| GPT-4-Turbo [37] | 0.71 | 2.57 | **2.55** | 1.49 | **2.57** | 2.65 |
| GPT-4o [37] | **0.86** | **2.68** | 2.46 | **1.78** | 2.45 | **2.68** |
| Avg. by models | 0.49 | 2.37 | 1.97 | 0.99 | 2.02 | 1.98 |

**Table 4: Detailed evaluation on Procedural Planning**, including both high-level and middle-level planning. Each level is evaluated across three types of query formulation as discussed in § 3.2 (*i.e.,*vision, text, and vision & text). Columns highlighted with colors are the primary evaluation settings. The maximum score is 5.

**Procedural planning.** Tab. 4 studies the impact of different query formulations for planning. On both high and middle-level: **(i)** The vision-only setting is significantly challenging (especially for high-level, 0.49 versus 2.37 for textual). Among the models, GPT-4o demonstrates the strongest visual reasoning ability. **(ii)** All models, except CogAgent [35]with a small LLM [46], exhibit similar performance on textual-only inputs, as the textual query concretely indicates the key operations or effects type. This suggests that if we have clear and detailed textual instructions, a text LLM may be sufficient for this stage. **(iii)** We do not observe a significant gain in the vision+text setting compared to text-only, which requires strong interleaved UI perception abilities.

**Action executions.** Tab. 5 examines the impact of different atomic actions on model performance. We summarize our findings as below. **(i) Click:** We prompt multi-modal LLMs to output coordinates

| Model | Grd.? | 1. Click | | 2. Drag | | 3. Type / Press | | 4. Scroll | Action |
|---|---|---|---|---|---|---|---|---|---|
| | | Dist. ↓ | Recall ↑ | Dist. ↓ | Recall ↑ | Recall | Prec. | Acc. | Full |
| Random | – | 49.9 | 0.7 | 47.2 | 0.0 | – | – | 31.3 | 8.0 |
| *LLMs* | | | | | | | | | |
| LLama3-70B [44] | – | – | – | – | – | 84.9 | 81.3 | – | 20.3 |
| Mixtral-8x22B [45] | – | – | – | – | – | 82.6 | 78.5 | – | 19.6 |
| GPT-3.5-Turbo [43] | – | – | – | – | – | **93.1** | **89.5** | – | 22.4 |
| *Multi-modal LLMs* | | | | | | | | | |
| CogAgent [35] | ✓ | 30.9 | 3.4 | 44.7 | 0.0 | – | – | 26.6 | 7.5 |
| Qwen-VL-Max [42] | ✓ | 46.8 | 0.0 | 42.0 | 0.3 | 84.3 | 73.0 | 42.2 | 28.9 |
| Gemini-Pro-Vision [41] | | 40.7 | 5.0 | 40.8 | 0.0 | 86.4 | 82.2 | 7.8 | 23.8 |
| Claude-3-Opus [40] | | 30.7 | 7.0 | 30.6 | 1.7 | 92.5 | 88.1 | 60.9 | 39.4 |
| GPT-4-Turbo [37] | | 23.8 | 10.0 | 31.3 | 1.4 | 92.3 | 88.8 | 37.5 | 34.4 |
| GPT-4o [37] | | 16.6 | 17.7 | 21.9 | 2.5 | 92.3 | 89.0 | 81.3 | 47.6 |
| *Modular methods: LLMs + Tools* | | | | | | | | | |
| GPT-3.5 + OCR [43] | ✓ | 16.8 | 48.7 | 36.4 | 5.5 | 93.1 | 89.5 | 56.3 | 50.0 |
| GPT-4T + OCR [43] | ✓ | 14.8 | 55.1 | 26.6 | 12.2 | 92.3 | 88.8 | 40.6 | 49.2 |
| GPT-4o + OCR [43] | ✓ | **12.0** | **60.1** (+42.4) | 25.7 | **11.3** (+8.8) | 92.3 | 89.0 | 82.8 (+1.5) | **56.3** (+8.7) |
| GPT-4T + SoM [43] | ✓ | 19.1 | 30.6 | 25.3 | 4.1 | 92.3 | 88.8 | 53.1 | 44.2 |
| GPT-4o + SoM [32] | ✓ | 15.7 | 35.9 (+18.2) | **22.9** | 3.0 (+0.5) | 92.3 | 89.0 | **89.0** (+7.7) | 54.3 (+6.7) |

**Table 5: Detailed evaluation on Actions Executions.** We report model performance on four types of atomic action execution. The full score is the sum of Click recall, Drag recall, Type precision, and Scroll accuracy. **Grd.** indicates whether the model has explicit grounding ability such as output element's coordinates. In the bottom half, we equip LLMs with tools like SoM [32] and OCR [47].

by providing screenshots with its resolutions, and we found that they can have a proper estimation, with meaningful improvement over random score but with poor recall. Notably, closed-source LLMs demonstrate better grounding abilities than grounding-based models such as CogAgent; Enhancing LLMs with tools such as OCR [47] or SoM [32] significantly improves model performance. Notably, for the text-based GPT-3.5 with OCR, it achieves a 48.7 recall. **(ii) Drag:** To perform Drag, it requires models to accurately localize the movement at both the start and end points. The best model, GPT4-o with OCR, yields only 11.3 recall. For LLMs with tools, OCR brings 8.8 recall gain over the base model, which is even more helpful than SoM as it helps to precisely localize text for the button, while SoM often suffers from poor segmentation results. **(iii) Type / Press:** Regarding keyboard activity, most models achieve good scores, as large-scale instruction-tuned LLMs generally is decent at coding, making the LLMs even more competent for this task. **(iv) Scroll:** For Scroll, models must infer not only whether an element appears but also its order relative to other elements. GPT-4o is the top-performing model, while Gemini scores extremely low, often preferring outputs without scrolling.

## 4.3 Performance by Task Difficulty.

**High planning by different software.** Fig. 5 (top) shows mid-level plan scores (visual query) across different software. Models perform highest on Powerpoint, which is more commonly used than others. On æ and Photoshop, model performance drops significantly as they are professional software. It is worth mentioning that being web-based, Runway and Stable Diffusion remains challenging because these novel applications are relatively new to the MLLMs.

**Middle planning by action number.** Fig. 5 (bottom) shows the mid-level planning scores (visual + text query) by the number of actions per task. Scores tend to decrease as the number of actions increases, demonstrating the difficulty of long procedural GUI tasks.

## 4.4 Qualitative Results

In Fig. 6, we visualize model performance and failure cases. In (a) High-level planning, GPT-4o and Gemini-Pro-V successfully predict the sub-tasks for the slide with the 3D model. GPT-4o also accurately identifies the Morph transition effect, achieving the best score. In (b) Mid-level planning, both models inserted and adjusted the 3D Shiba Inu model. However, Gemini-Pro-V introduces unnecessary operations,

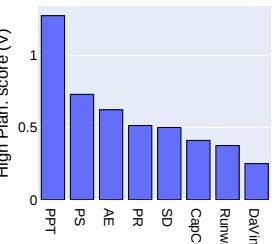

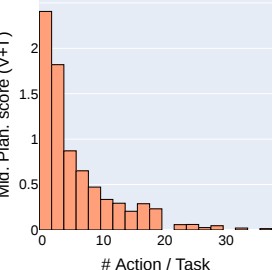

**Figure 5: Top:** High plan. score (V) by diff. software; **Bottom:** Mid. plan score (V+T) by action number.

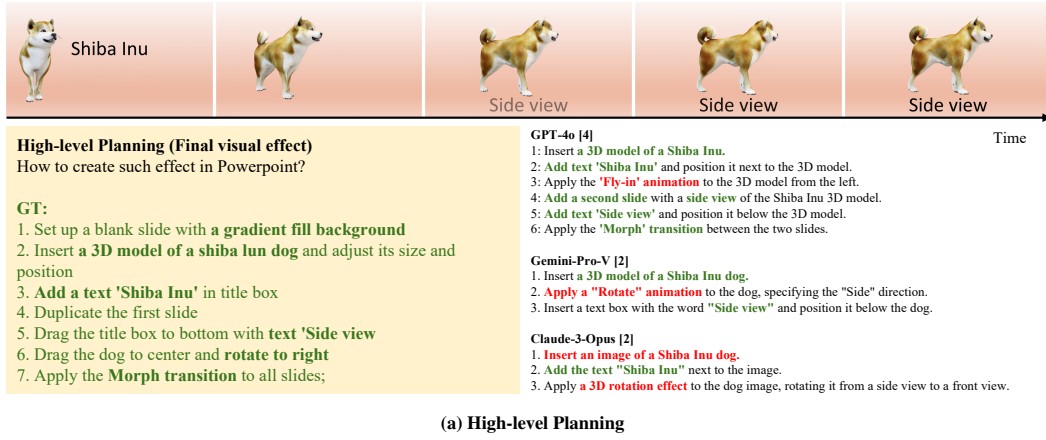

**(a) High-level Planning**

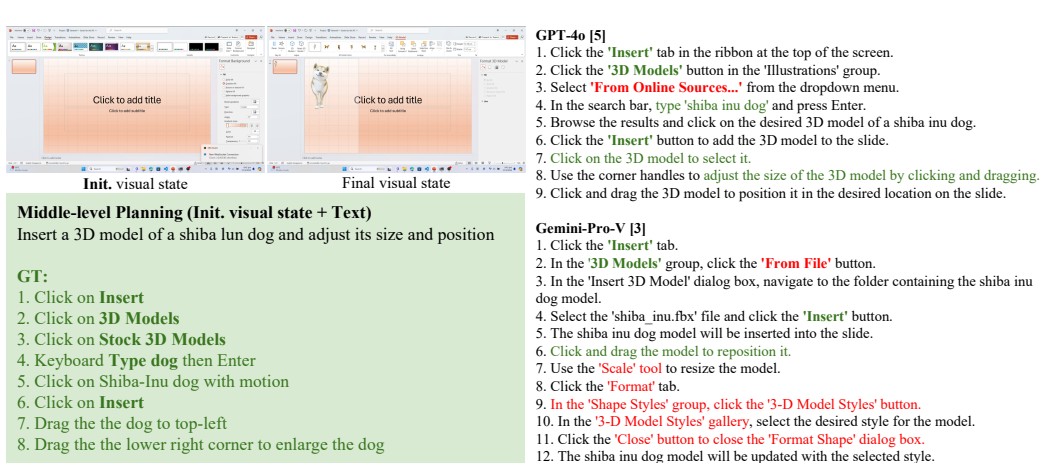

**(b) Middle-level Planning**

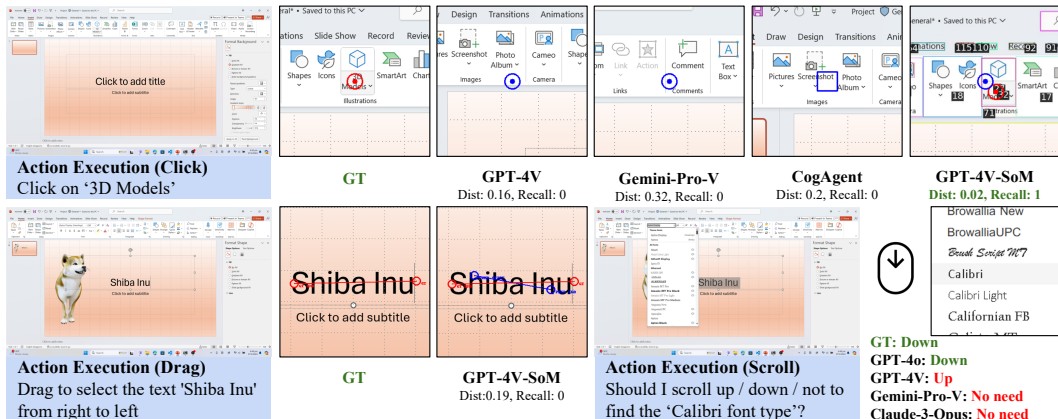

**(c) Atomic Execution** with Click, Drag and Scroll.

Figure 6: **Qualitative Results on VideoGUI with Powerpoint software**. The color **green** indicates the human references (GT), while **red** indicates wrong model predictions.

such as shape styles and formatting, leading to discrepancies in positioning and scaling. In (c) Atomic execution, models are assessed on precise actions. In Drag, GPT-4V selects part of the letters, but as the pixel distance is larger than the threshold, it still receives a recall of 0.

## 4.5 Simulator Experiments

To simulate the real application scenario, we use the best performing LLM GPT-4o and build a simple agent baseline as shown in Fig. 7. We evaluate this agent on the most popular software (Powerpoint) to study its behavior.

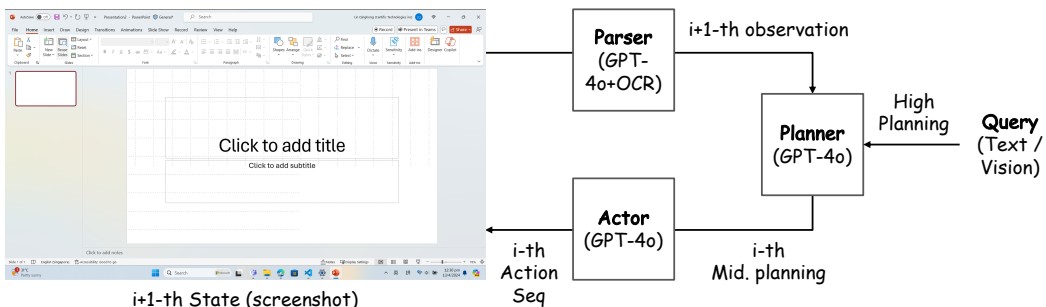

**Figure 7: Our Minimalist GUI Agent Framework** consists of three components: a Parser, a Planner, and an Actor. The Planner receives input queries, which may be either vision previews or text instructions. It then conducts high-level planning and generates mid-level plans for the Actor. The Actor executes these plans by performing a sequence of actions. After action execution, the current state (screenshot) is captured and sent back to the Parser to gather observations. These observations are then relayed to the Planner for subsequent planning.

| Model | Settings | VideoGUI Eval. | | | Full task Eval. | |
|---|---|---|---|---|---|---|
| | | High Plan. | Mid Plan. | Action | Success Rate | Rank (Arena) ↓ |
| GUI Agent w/ GPT-4o [37] | Orig. Query (V) | 17.1 | 53.5 | 56.3 | 0 | 2.50 |
| | w. GT High Plan. | 100.0 | 53.5 | 56.3 | 0 | 1.88 |
| | w. GT High & Mid Plan. | 100.0 | 100.0 | 56.3 | 0 | **1.38** |

**Table 6: Simulator Evaluation on VideoGUI's PPT *full tasks*.**

Tab. 6 presents the model performance on full task execution in our simulator environment. We see that completing the full task is extremely challenging for the GPT4o agent, with a notable 0 success rate for all variants. This again supports the design of our hierarchical evaluation, as the zero success rate simply implies the model/agent fail to execute the full task, without enough information in where they succeed or fail, or even how these models/agents perform relatively to each other. Therefore, we introduce another metric, Rank (Arena), which compares the final outcome of their execution. Specifically, we ask human to perform manual inspection, and rank the comparing models by the similarities between the final results and the GT. We found that when injected with GT planning (both high or mid.-level), the full-task execution can be significantly improved. These results echoes our observations of low model performance in high-level and mid-level planning in the main paper, which are the bottlenecks of successful full-task executions.

We visualize the final outcome of the three agent variants in Fig. 9 and Fig. 11.

| Model | Settings | VideoGUI Eval. | | Subtask Eval. | |
|---|---|---|---|---|---|
| | | Mid Plan. | Action | Success Rate (%) | Avg. Round ↓ |
| GUI Agent w/ GPT-4o [37] | Orig. Query (V+T) | 53.5 | 56.3 | 20.0 | 5.4 |
| | w. GT Mid Plan. | 100 | 56.3 | **50.0** | **3.3** |

**Table 7: Simulator Evaluation on VideoGUI's PPT *subtasks*.**

In Tab. 7, we examine the performance of the GPT-4o agent in subtask competitions. Since subtasks do not necessitate high-level planning, we primarily investigate two variants: one with and one without manually provided middle-level planning, referred to as action sequences. Our study yields two key findings: (*i*) Despite the simplicity of these tasks, the original GPT-4o agent achieves a success rate of only 20.0%. With the assistance of manual plans, there is a 30% increase in success rate. (*ii*) For simple subtasks, the agent typically requires more extensive procedural execution compared to manual demonstrations (+2.1), which often represent the optimal pathway. This redundancy is exacerbated in complex tasks. Therefore, enhancing planning capabilities is essential for achieving efficient system with accurate success rates.

## 5 Conclusion

In this work, we introduced VideoGUI, a multi-modal benchmark for advanced GUI tasks sourced from high-quality instructional videos targeting professional and novel software. VideoGUI, with its long procedural tasks, hierarchical manual annotations, and well-established evaluation metrics, provides clear signals for existing limitations and areas for improvement. By comparing state-of-the-art models, we highlight the challenges of visual-oriented GUI automation and the potential of instructional videos for advancing GUI task automation.

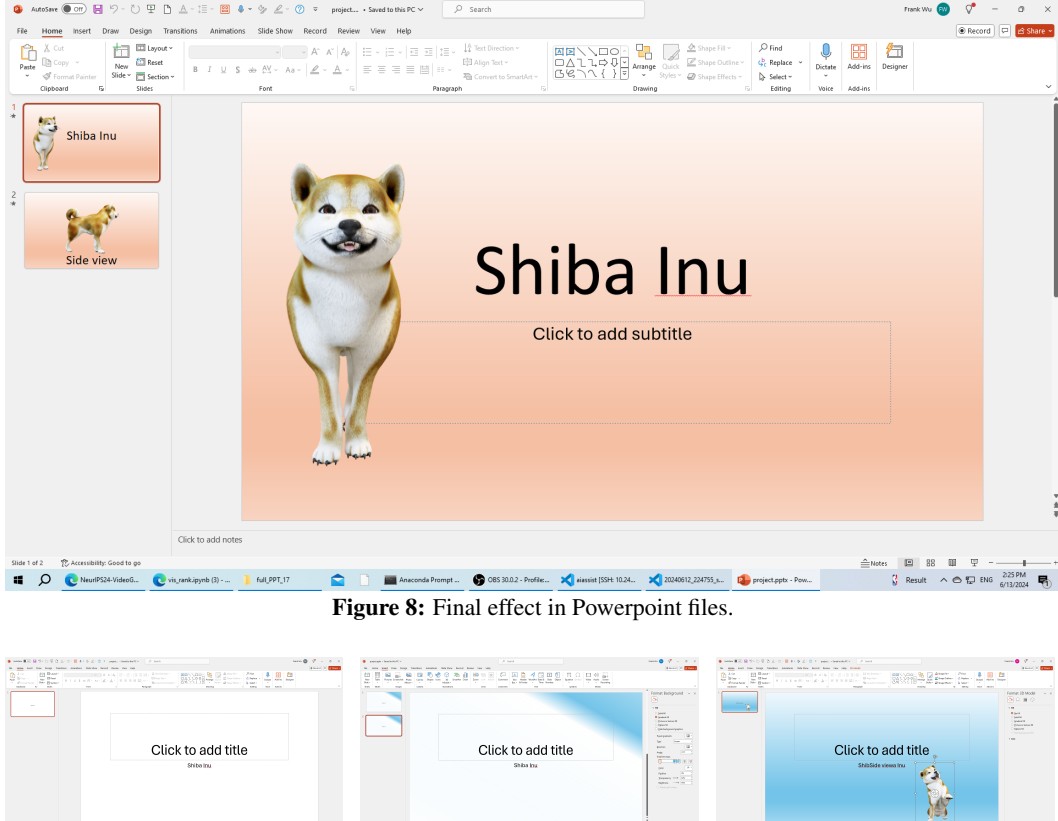

**Figure 8:** Final effect in Powerpoint files.

| (a) GPT-4o | (b) GPT-4o w. GT High Plan | (c) GPT-4o w. GT High+Mid. Plan |
| --- | --- | --- |

**Figure 9: Example of final outcome with our simple GPT-4o agent in simulated environment.** When provided with GT planning (c), the GUI agent successfully inserts the 3D model. However, it still fails to match the background color.

**Acknowledgement** This research is supported by the National Research Foundation, Singapore under its AI Singapore Programme (AISG Award No: AISG3-RP-2022-030).

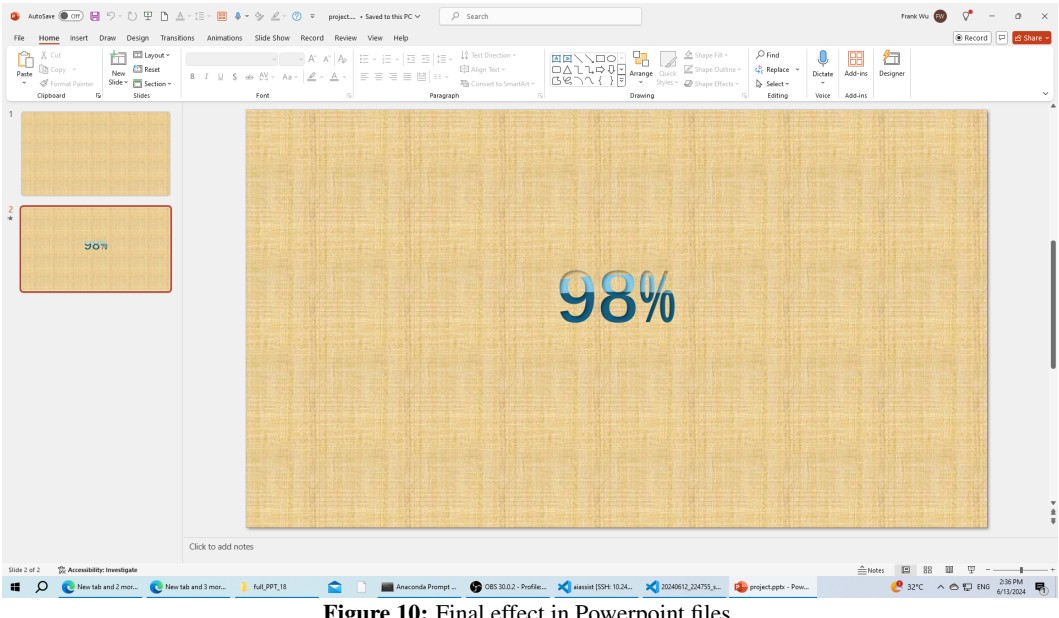

**Figure 10:** Final effect in Powerpoint files.

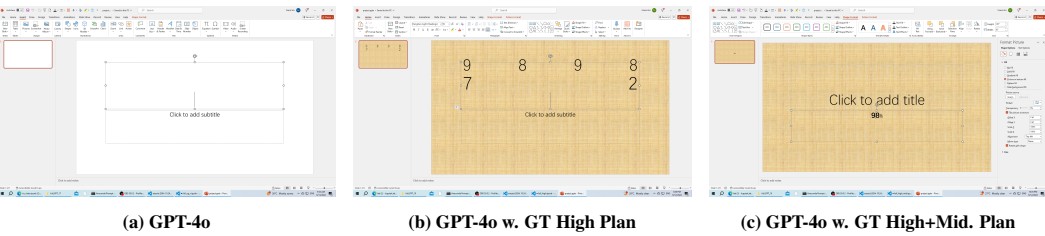

(a) GPT-4o

(b) GPT-4o w. GT High Plan

(c) GPT-4o w. GT High+Mid. Plan

**Figure 11: Example of final outcome with our simple GPT-4o agent in simulated environment.** Guided by the GT planning, both (b) and (c) successfully insert the textual background, while the (c) can accurately type '98%'.

# A  Experimental Settings

## A.1  Data Collection Settings

We use OBS Studio [48] software to record the demonstration videos and capture the user's screenshots. Notably, in the screenshots, the user's cursor is not recorded, which is beneficial as the screenshots can be used directly without revealing the target coordinates. We use pynput to monitor detailed user activity metadata, such as click location $[x, y]$, typed content, and scroll distance.

In Fig. 12, we display our manually labeled interface. Here, the annotator watches their key recording screenshots, with active regions such as the cursor coordinates highlighted in red. The annotators are then asked to enter the element name (e.g., "Drop-down menu of font color").

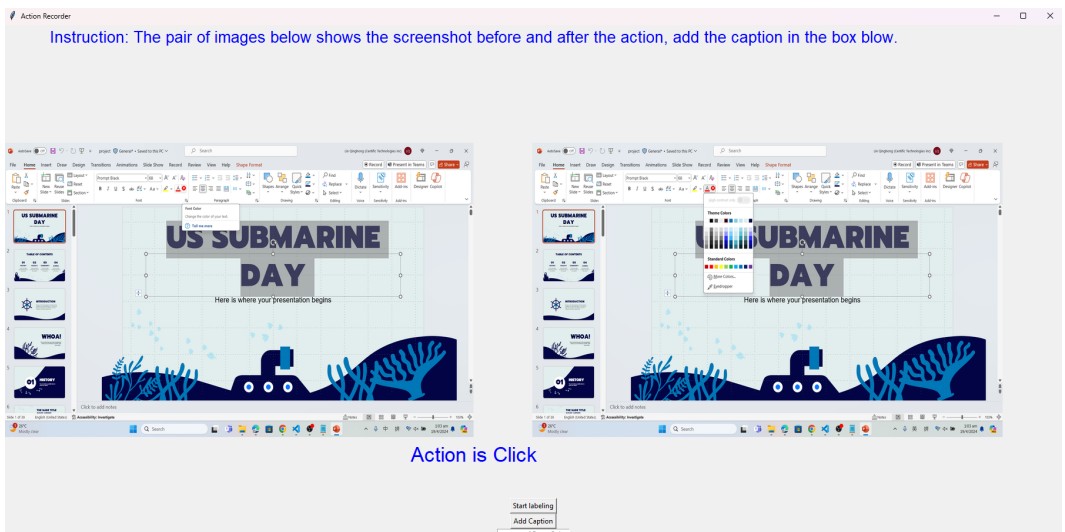

**Figure 12:** Illustration of Manual annotation tools. The user are asked to watch their keyframe in their recording, and prompt to provide the element name regarding action.

**Definition of Action Narration** (e.g., Drag)

We ask annotators to provide a textual quadruple for each drag action: [start position, end position, element, purpose]. The narration follows the format: Drag the [element] from [start position] to [end position] to [purpose]. Here, the start and end positions guide the annotators in identifying locations within the screenshot, with the element representing the object being dragged and the purpose defining the goal of the action (mainly movement or resizing)

For *movement*, the start point is usually the element's original parent element (or "original position" if unspecified), while the end point is determined by the target parent element (e.g., a panel). For *resizing*, the start point is based on the specific part of the element being dragged (e.g., "top-left corner of the circle"), and the end point is identified by relevant elements in the screenshot. Additionally, our focus is on making predictions that closely approximate the intended points, rather than matching exact coordinates, which is why the narration serves as an effective guide.

**Principle for selecting instruction videos.**

We selected the videos based on both topic relevance and quality:

**By Topic:** (i) Videos introducing novel concepts or features with visual preview effects, primarily for visual creation and editing software. (ii) Videos offering advanced knowledge beyond basic usage, such as "Top tips" for VLC Player.

**By Quality:** (iii) High-resolution videos with clear, step-by-step instructions; (iv) High-quality, accessible transcripts that users can easily follow.

## A.2 Baseline Details

| Model | Ref. link | Version (*e.g.,* model id) |
|---|---|---|
| LLama3-70B [44] | `deepinfra` | `meta-llama/Meta-Llama-3-70B-Instruct` |
| Mixtral-8x22B [45] | `deepinfra` | `mistralai/Mixtral-8x22B-Instruct-v0.1` |
| GPT-3.5-Turbo [43] | `OpenAI` | `gpt-3.5-turbo` |
| CogAgent [35] | `CogAgent` | `CogAgent-18B` |
| Qwen-VL-Max [42] | `Aliyun` | `qwen-vl-max` |
| Claude-3-Opus [40] | `Anthropic` | `claude-3-opus-20240229` |
| Gemini-Pro-V [41] | `Google` | `gemini-pro-vision` |
| GPT-4-Turbo [37] | `OpenAI` | `gpt-4-turbo` |
| GPT-4o [37] | `OpenAI` | `gpt-4o` |

## A.3 Evaluation Settings

**Click.** We detail how we calculate the distance metric. Assume we have a ground-truth point $[x_o, y_o]$ while the screenshot size is $H \times W$.

• If the model prediction is a bounding box $[x_1, y_1, x_2, y_2]$ (*e.g.,*CogAgent [35] or Qwen-VL-Max [42]):

We cannot only take the center of the bounding box as the click target for evaluation because it does not account for the area of the bounding box. As illustrated in Fig. 13 (a), if the center point is very close to the ground truth but the bounding box cover a large area, the distance between the center point and the groundtruth would be small. Therefore, we design our metric to penalize for the area of the bounding box. Specifically, we calculate the distance between the ground truth and the four corners of the bounding box and then take the average. For the predicted bounding box, the average distance $d$ is calculated as follows:

$$d = \frac{1}{4} \left( \sqrt{(x_o - x_1)^2 + (y_o - y_1)^2} + \sqrt{(x_o - x_1)^2 + (y_o - y_2)^2} \right.$$
$$\left. + \sqrt{(x_o - x_2)^2 + (y_o - y_1)^2} + \sqrt{(x_o - x_2)^2 + (y_o - y_2)^2} \right)$$

• If the model prediction is a coordinate $[x_1, y_1]$ (*e.g.,*as in GPT4V+SoM [32]):

We directly adopt the distance $d$ calculated by:

$$d = \sqrt{(x_o - x_1)^2 + (y_o - y_1)^2}$$

To normalize the pixel-level distance $d$ to $0 - 1$, a simple way is to divide $d$ by the maximum length in the screenshot, such as $\sqrt{H^2 + W^2}$. But in practice, the maximum length should be the distance between the ground-truth point and the farthest vertices, so we use that for normalization. The comparison between the two normalization methods is illustrated in Fig. 13 (b).

**Drag.** Drag is a combination of Clicks, so we simply adopt the click metric for the start and end point of drag, and take the average. The score is calculated as Dist $:= \frac{1}{2} \left( \frac{d_s}{D_s} + \frac{d_e}{D_e} \right)$ where $d_s$ is the pixel difference between predict start and GT start, while $D_s$ is the farthest vertices for the GT start; $d_e$ is the pixel difference between predict end and GT end, while $D_e$ is the farthest vertices for the GT end;

For Recall, it is calculated by:

$$\text{Recall (start, end)} = \begin{cases} 1 & if \quad \text{Recall (start)} \,\&\, \text{Recall (end)} \\ 0 & otherwise \end{cases}$$

**Type / Press.** For type/press, we evaluates whether the model can generate correct and efficient code to control keyboard activity. First, we prompt LLMs to write code for typing activity, and then we use `pynput` to monitor the keyboard outputs by executing the code. In Fig. 14, we show the pipeline for evaluating type/press activity. The model must generate the correct actions (e.g., `Ctrl+F`) with high precision, avoiding unnecessary actions such as redundant `Ctrl` presses.

**Scroll.** Fig. 15 illustrates how we construction QA pairs to evaluate on scroll action. Before scrolling, the target element is assumed to be outside of the visible area, prompting for a scroll action. After

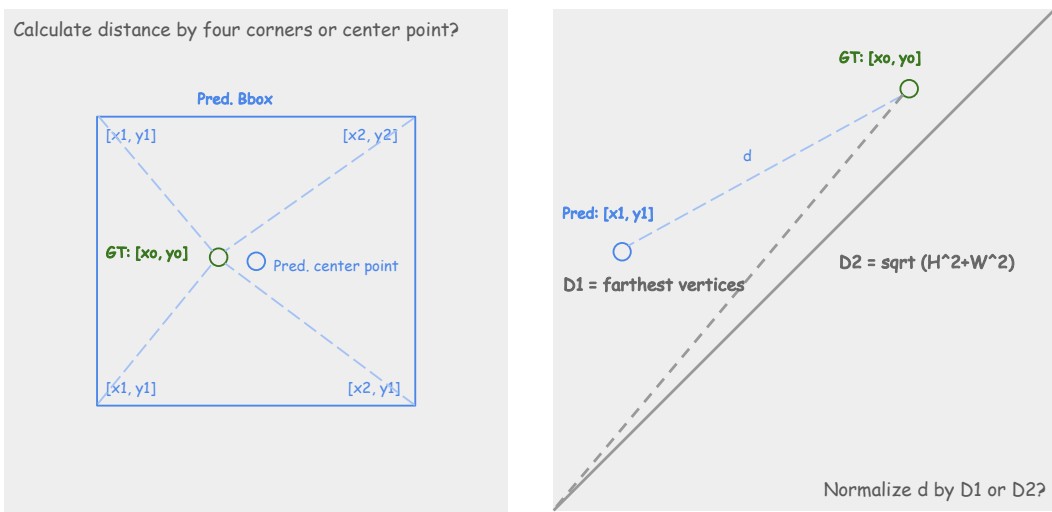

Figure 13: **(a) Illustration of why taking the distance btween the center point of a bounding box and groundtruth is not a proper measure of model performance on click.** As shown, the predicted bounding box center point is quite close to the ground-truth point, but the predicted bounding box area is large. **(b) Illustration of distance normalization.** To normalize the distance $d$ to $0-1$, a more proper term should be $D1$ (farthest vertices) rather than $D2$.

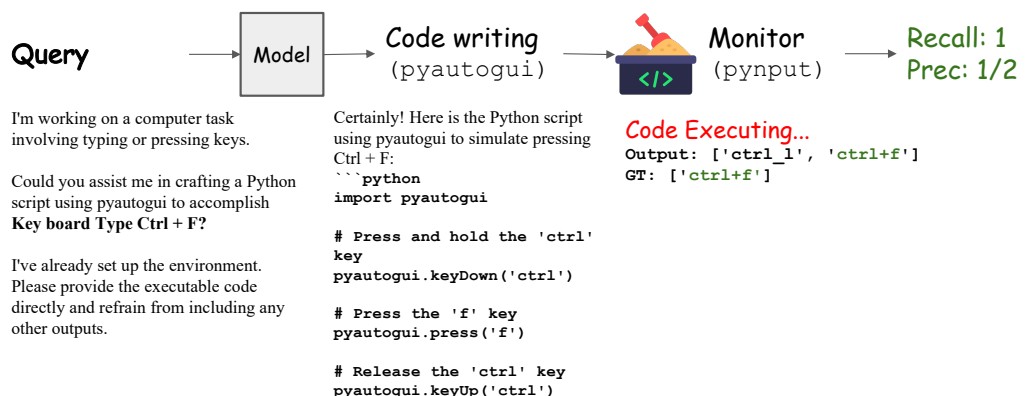

Figure 14: Illustration of how we evaluate the key / press action.

scrolling, the target element is assumed to be within the visible area, ready for the next action (*e.g.,*Click shown in the figure). Thereby, we can construct the QA pairs under these assumptions.

For each scroll, we create two QA pairs with the following GT answers: "scroll (up/down)" for the screenshot before scrolling and "no" for the screenshot after scrolling. We randomly shuffle the order of answer options to make the final testing samples.

# B  More Discussion

| | Pros | Cons | Environment |
|---|---|---|---|
| **Human-only** | High-quality, Interpretability | Extremely cost | Real Simulator |
| **LLM-only** | Fully automatic | Hallucinations, might be unreliable | No required |
| **Human & LLM (Human anno. + LLM judge)** | Sufficient signals for each stage; Automatic once we collected all annotations | Require annotations for each task in advance. | No required |
| **Human & LLM (Agent exec. + Human verify)** | Check whether agent indeed complete the full-task | Require human check output (but is fast) | Real Simulator |

**Table 8:** Comparison between Human evaluation and LLM as Judge.

**Human v.s. LLMs as Judge.** In evaluation, accuracy is the most important. As shown in the below Table, while Human-only annotation ensures high-quality results, it is extremely time-consuming. On

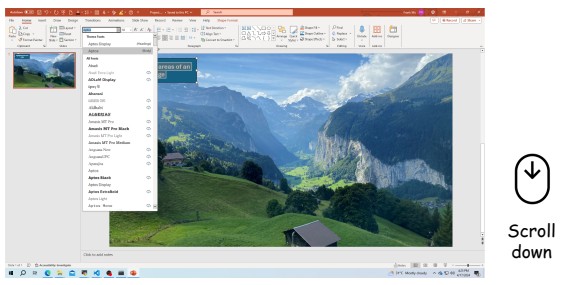

Before Scrolling, we assume that
**[the target element]** is **not within** the screenshot.

After Scrolling, we assume that
**[the target element]** should **be inside** the screenshot.

**Next Action:** Click **[the maiandra GD button]**

**Question: Should I scroll to find the maiandra GD button?**
**Answer:** Scroll down

**Question: Should I scroll to find the maiandra GD button?**
**Answer:** No

**Figure 15:** Illustration of how we create the scroll QA pair.

the other hand, LLM-only allows for full automation but may lead to issues such as hallucinations, rendering the output potentially unreliable. Consequently, a hybrid Human+LLM combining human expertise and LLMs is a reasonable compromise, offering a balanced solution.

Besides, in Table row 3 i.e., after obtaining annotations for each task, our main evaluation pipeline actually doesn't need human assistance, and it is an acceptable automatic solution. Nevertheless, there are opportunities for further refinement. For instance, we could develop task-specific pipelines tailored to individual outcomes. This could involve feeding the final output generated by the agent into a verification process, which applies specific rules to assess whether the expected content (e.g., text, shapes) is present at the desired location.

**Consideration of Alternative way during Evaluation.** There might be multiple action trajectories to accomplish one task shown in instructional videos. However, it is important to note that *these videos, typically created by experts, often demonstrate the most common and representative approach to completing a task.*

Moreover, we provide several strategies:

1. Firstly, we need to clarify that the goal of high-level planning is for recognizing the key milestone (e.g., particular animation type) rather than detailed action sequence. So low high-level plan scores are mainly due to models failing to recognize rather than alternative pathways. The alternative issue is mainly focused on middle-level planning.

2. To address this at the planning stage, We incorporate the LLM Critic to account for human-like subjective reasoning. This method is capable of considering alternative actions, such as recognizing that `Ctrl + C` is equivalent to Right-click + Copy.

3. To allow any alternative planning possibilities as long as the model successfully achieves the final outcome, we include the metric of Success Rate, as demonstrated in Supp. Tab. 13-14. This evaluation approach inherently supports alternative planning trajectories.

4. A possible future effort is that we provide diverse enough planning annotations by multiple annotators for the same task.

**Future extension.** So far we focus on Windows, but software versions among different platforms might bring differences. Extend to cross-software evaluation, such as first collecting video assets on a website then editing them in PR / AE.

## C   Benchmark Statistics

**Software distributions** In Tab. 9, we present the software distribution on VideoGUI.

**Manual Recording Cost.** In Fig. 16a, we present the screenshot resolution distribution primarily used for action execution.

| Software | Platform | # Full Task | # Subtask | # Action per full task | # Action per subtask |
|---|---|---|---|---|---|
| Powerpoint | Windows | 8 | 52 | 47.6 | 8.5 |
| StableDiffusion | Web + Windows | 10 | 69 | 19.0 | 4.0 |
| Runway | Web | 11 | 63 | 24.3 | 4.7 |
| Photoshop | Windows | 10 | 69 | 19.0 | 4.0 |
| After Effects | Windows | 13 | 67 | 29.3 | 7.2 |
| Premiere Pro | Windows | 7 | 38 | 15.4 | 4.5 |
| Capcut | Web + Windows | 10 | 46 | 9.4 | 3.6 |
| DaVinci | Windows | 11 | 44 | 18.8 | 4.7 |
| YouTube | Web | 0 | 13 | 0 | 4.3 |
| Web Stock | Web | 0 | 12 | 0 | 9.7 |
| VLC player | Windows | 0 | 12 | 0 | 9.2 |
| Total | – | 82 | 463 | 23.7 | 5.8 |

**Table 9:** VideoGUI's software distribution.

**Screenshot's resolutions.** In Fig. 16b, we present the distribution of manual recording time per subtask, with an average of 55 sec.

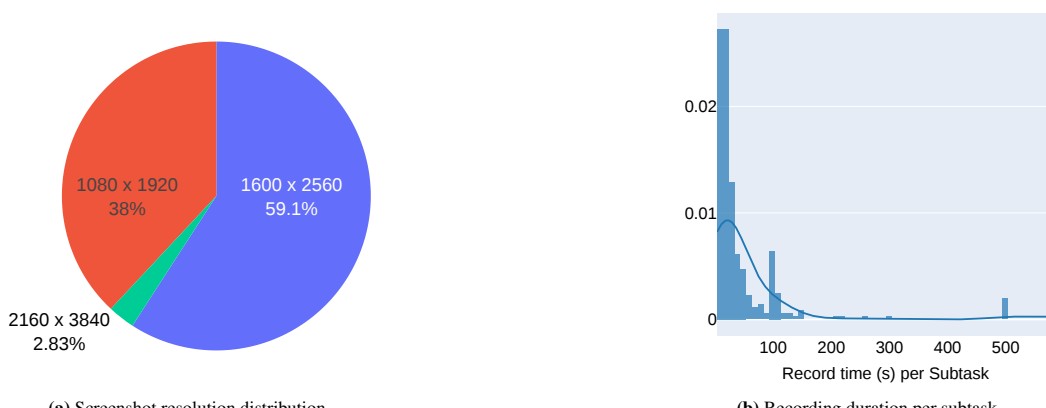

(a) Screenshot resolution distribution.

(b) Recording duration per subtask.

**Figure 16:** Distribution of **(a) Screenshot resolution** and **(b) Human recording time.**

**World Cloud.** In Fig.17, we present VideoGUI's Word Cloud, where the most frequent words are atomic actions (*e.g.,* click, drag, type) and commonly used proper nouns (*e.g.,*, layer, background, pannel) in the GUI.

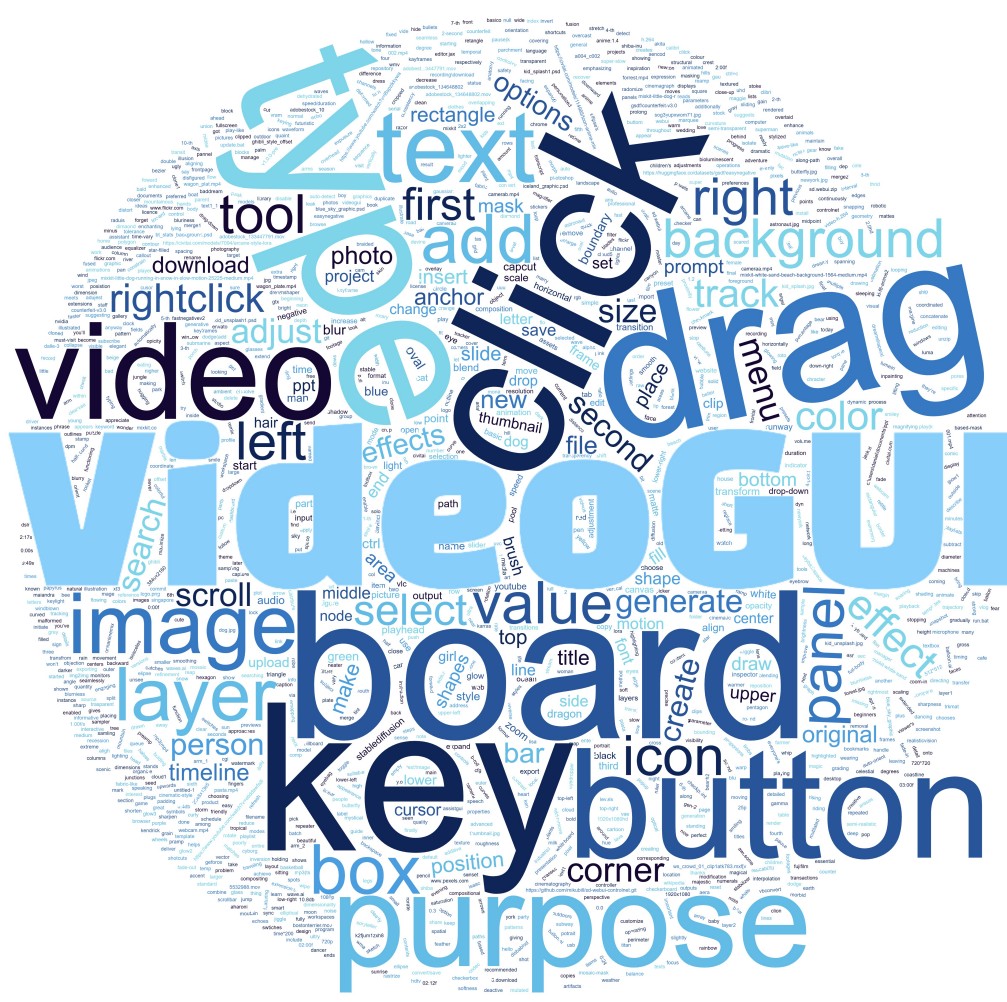

**Figure 17:** VideoGUI World Clouds

# D   Dataset Examples

**Data samples.** In this section, we display the visual-preview data samples, which are mainly focused on visual creation or editing.

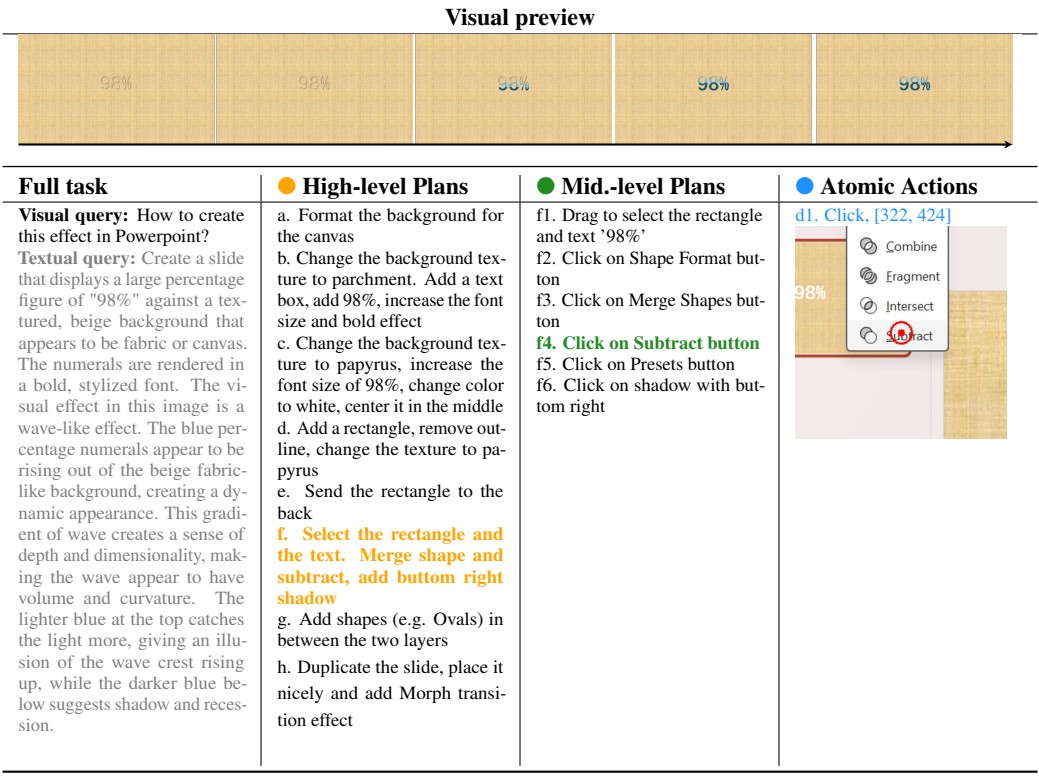

| Full task | 🟡 High-level Plans | 🟢 Mid.-level Plans | 🔵 Atomic Actions |
|---|---|---|---|
| **Visual query:** How to create this effect in Powerpoint? **Textual query:** Create a slide that displays a large percentage figure of "98%" against a textured, beige background that appears to be fabric or canvas. The numerals are rendered in a bold, stylized font. The visual effect in this image is a wave-like effect. The blue percentage numerals appear to be rising out of the beige fabric-like background, creating a dynamic appearance. This gradient of wave creates a sense of depth and dimensionality, making the wave appear to have volume and curvature. The lighter blue at the top catches the light more, giving an illusion of the wave crest rising up, while the darker blue below suggests shadow and recession. | a. Format the background for the canvas  b. Change the background texture to parchment. Add a text box, add 98%, increase the font size and bold effect  c. Change the background texture to papyrus, increase the font size of 98%, change color to white, center it in the middle  d. Add a rectangle, remove outline, change the texture to papyrus  e. Send the rectangle to the back  **f. Select the rectangle and the text. Merge shape and subtract, add buttom right shadow**  g. Add shapes (e.g. Ovals) in between the two layers  h. Duplicate the slide, place it nicely and add Morph transition effect | f1. Drag to select the rectangle and text '98%'  f2. Click on Shape Format button  f3. Click on Merge Shapes button  **f4. Click on Subtract button**  f5. Click on Presets button  f6. Click on shadow with buttom right | d1. Click, [322, 424] |

**Table 10:** Video Creation (*i.e.,* animation) example with **Powerpoint.**

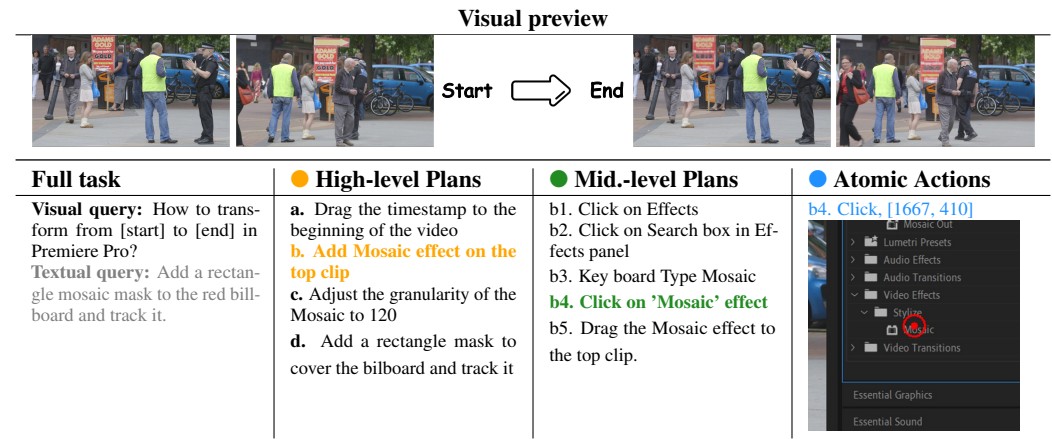

| Full task | 🟡 High-level Plans | 🟢 Mid.-level Plans | 🔵 Atomic Actions |
|---|---|---|---|
| **Visual query:** How to transform from [start] to [end] in Premiere Pro? **Textual query:** Add a rectangle mosaic mask to the red billboard and track it. | **a.** Drag the timestamp to the beginning of the video  **b. Add Mosaic effect on the top clip**  **c.** Adjust the granularity of the Mosaic to 120  **d.** Add a rectangle mask to cover the bilboard and track it | b1. Click on Effects  b2. Click on Search box in Effects panel  b3. Key board Type Mosaic  **b4. Click on 'Mosaic' effect**  b5. Drag the Mosaic effect to the top clip. | b4. Click, [1667, 410] |

**Table 11:** Video Editing example with **Premiere Pro.**

| Visual preview | | | |
|---|---|---|---|

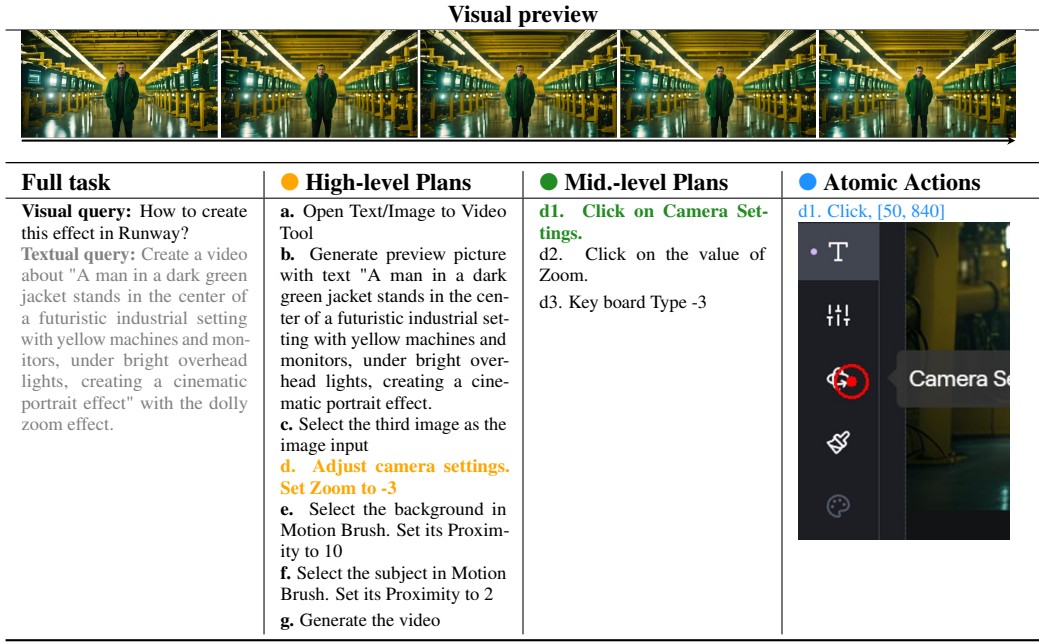

| Full task | ● High-level Plans | ● Mid.-level Plans | ● Atomic Actions |
|---|---|---|---|
| **Visual query:** How to create this effect in Runway? **Textual query:** Create a video about "A man in a dark green jacket stands in the center of a futuristic industrial setting with yellow machines and monitors, under bright overhead lights, creating a cinematic portrait effect" with the dolly zoom effect. | **a.** Open Text/Image to Video Tool **b.** Generate preview picture with text "A man in a dark green jacket stands in the center of a futuristic industrial setting with yellow machines and monitors, under bright overhead lights, creating a cinematic portrait effect." **c.** Select the third image as the image input **d. Adjust camera settings. Set Zoom to -3** **e.** Select the background in Motion Brush. Set its Proximity to 10 **f.** Select the subject in Motion Brush. Set its Proximity to 2 **g.** Generate the video | **d1. Click on Camera Settings.** d2. Click on the value of Zoom. d3. Key board Type -3 | d1. Click, [50, 840] |

**Table 12:** Video Creation example with **Runway.**

| Visual preview | | | |
|---|---|---|---|

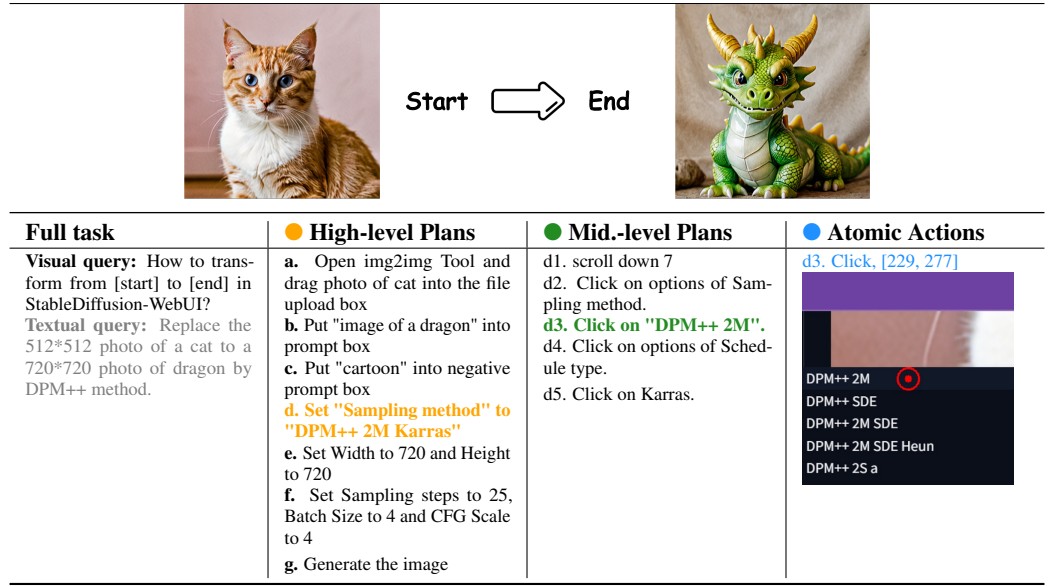

| Full task | ● High-level Plans | ● Mid.-level Plans | ● Atomic Actions |
|---|---|---|---|
| **Visual query:** How to transform from [start] to [end] in StableDiffusion-WebUI? **Textual query:** Replace the 512*512 photo of a cat to a 720*720 photo of dragon by DPM++ method. | **a.** Open img2img Tool and drag photo of cat into the file upload box **b.** Put "image of a dragon" into prompt box **c.** Put "cartoon" into negative prompt box **d. Set "Sampling method" to "DPM++ 2M Karras"** **e.** Set Width to 720 and Height to 720 **f.** Set Sampling steps to 25, Batch Size to 4 and CFG Scale to 4 **g.** Generate the image | d1. scroll down 7 d2. Click on options of Sampling method. **d3. Click on "DPM++ 2M".** d4. Click on options of Schedule type. d5. Click on Karras. | d3. Click, [229, 277] |

**Table 13:** Image Editing example with **StableDiffusion-WebUI.**

| Visual preview |
|:---:|

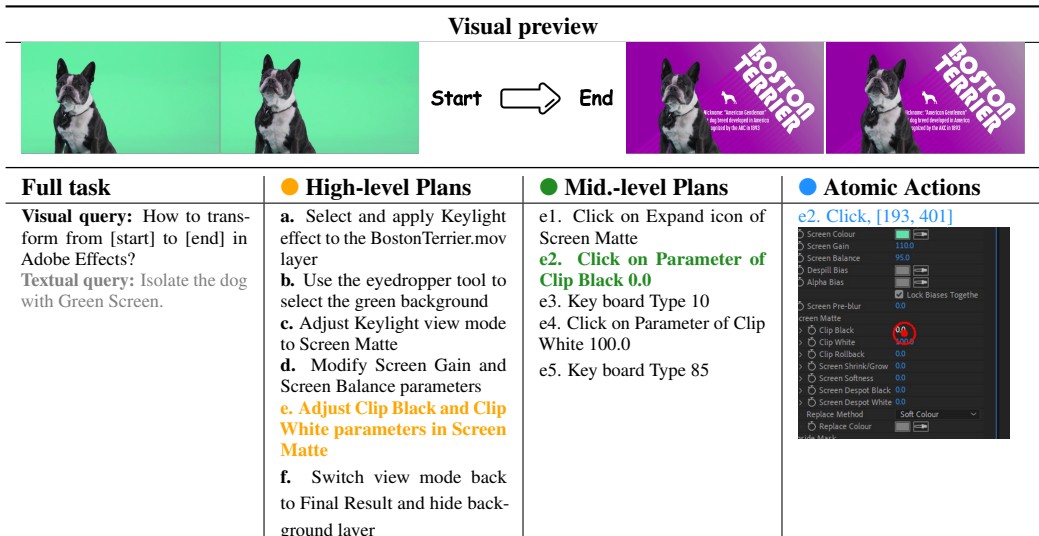

| Full task | 🟠 High-level Plans | 🟢 Mid.-level Plans | 🔵 Atomic Actions |
|---|---|---|---|
| **Visual query:** How to transform from [start] to [end] in Adobe Effects? **Textual query:** Isolate the dog with Green Screen. | **a.** Select and apply Keylight effect to the BostonTerrier.mov layer **b.** Use the eyedropper tool to select the green background **c.** Adjust Keylight view mode to Screen Matte **d.** Modify Screen Gain and Screen Balance parameters **e. Adjust Clip Black and Clip White parameters in Screen Matte** **f.** Switch view mode back to Final Result and hide background layer | e1. Click on Expand icon of Screen Matte **e2. Click on Parameter of Clip Black 0.0** e3. Key board Type 10 e4. Click on Parameter of Clip White 100.0 e5. Key board Type 85 | e2. Click, [193, 401] |

**Table 14:** Video Editing example with **Adobe Effects.**

| Visual preview |
|:---:|

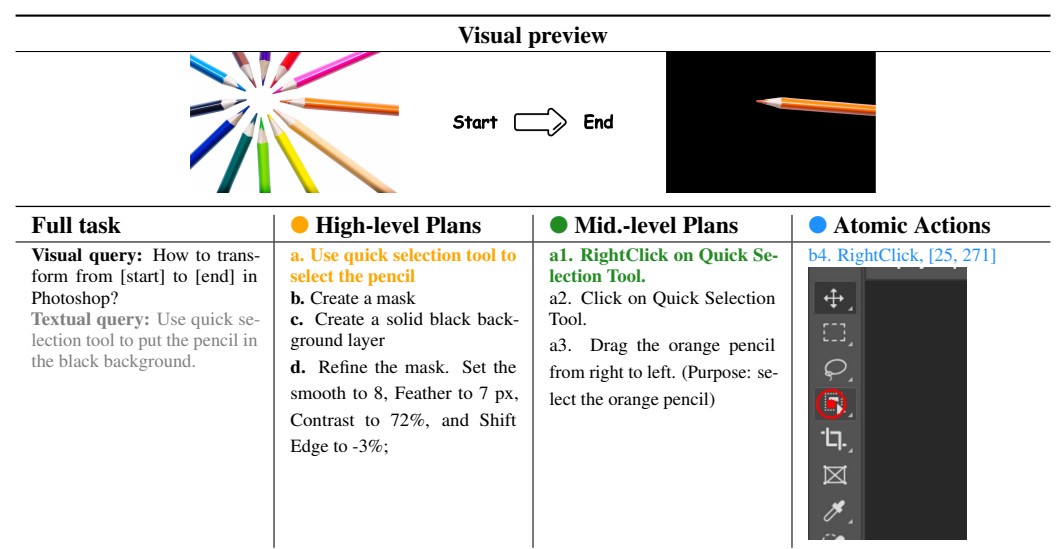

| Full task | 🟠 High-level Plans | 🟢 Mid.-level Plans | 🔵 Atomic Actions |
|---|---|---|---|
| **Visual query:** How to transform from [start] to [end] in Photoshop? **Textual query:** Use quick selection tool to put the pencil in the black background. | **a. Use quick selection tool to select the pencil** **b.** Create a mask **c.** Create a solid black background layer **d.** Refine the mask. Set the smooth to 8, Feather to 7 px, Contrast to 72%, and Shift Edge to -3%; | **a1. RightClick on Quick Selection Tool.** a2. Click on Quick Selection Tool. a3. Drag the orange pencil from right to left. (Purpose: select the orange pencil) | b4. RightClick, [25, 271] |

**Table 15:** Image Editing example with **Photoshop.**

**Visual preview**

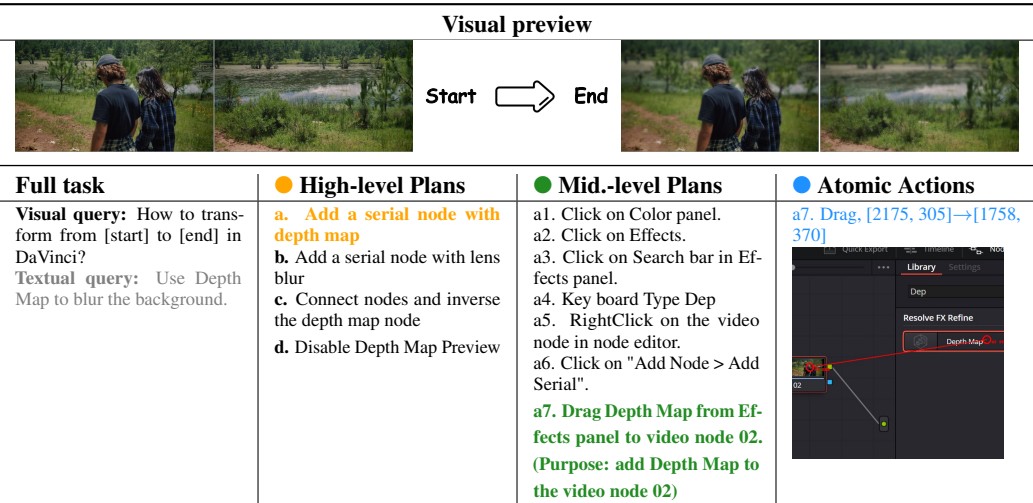

Start ⟹ End

| Full task | 🟠 High-level Plans | 🟢 Mid.-level Plans | 🔵 Atomic Actions |
|---|---|---|---|
| **Visual query:** How to transform from [start] to [end] in DaVinci? **Textual query:** Use Depth Map to blur the background. | **a. Add a serial node with depth map** **b.** Add a serial node with lens blur **c.** Connect nodes and inverse the depth map node **d.** Disable Depth Map Preview | a1. Click on Color panel. a2. Click on Effects. a3. Click on Search bar in Effects panel. a4. Key board Type Dep a5. RightClick on the video node in node editor. a6. Click on "Add Node > Add Serial". **a7. Drag Depth Map from Effects panel to video node 02. (Purpose: add Depth Map to the video node 02)** | a7. Drag, [2175, 305]→[1758, 370] |

**Table 16:** Video Editing example with **DaVinci.**

**Visual preview**

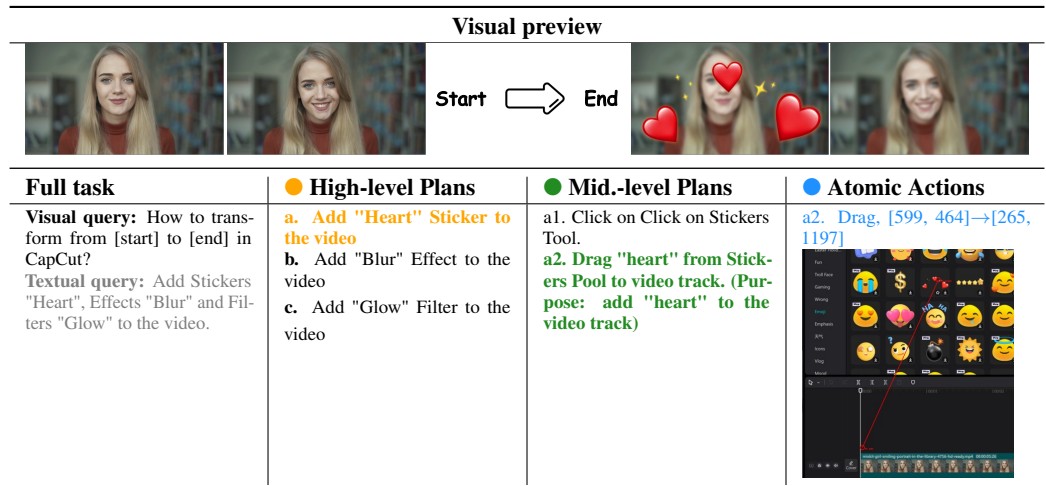

Start ⟹ End

| Full task | 🟠 High-level Plans | 🟢 Mid.-level Plans | 🔵 Atomic Actions |
|---|---|---|---|
| **Visual query:** How to transform from [start] to [end] in CapCut? **Textual query:** Add Stickers "Heart", Effects "Blur" and Filters "Glow" to the video. | **a. Add "Heart" Sticker to the video** **b.** Add "Blur" Effect to the video **c.** Add "Glow" Filter to the video | a1. Click on Click on Stickers Tool. **a2. Drag "heart" from Stickers Pool to video track. (Purpose: add "heart" to the video track)** | a2. Drag, [599, 464]→[265, 1197] |

**Table 17:** Video Editing example with **CapCut.**

# E  Prompts Templates

**Procedural Planning.** In Tab.18 and Tab.19, we present the prompt templates for high-level and mid-level planning, respectively. These templates are conditioned on the query formulation, such as whether the start or end visual effects are provided, or paired with the textual query.

```
def get_high_prompt(vis_start=True, vis_end=True, txt=None,
software=None):

  PROMPT = f"You are a software assistant professional at {software}."

  if vis_start and vis_end:
      PROMPT += "Given two sequence of image frames about the initial visual effect and
the final visual effect"
  elif vis_end:
      PROMPT += "Given a sequence of image frames about the final visual effect"
  else:
      PROMPT += " You are provided"

  if txt:
      PROMPT += " with a task textual description"

  PROMPT += " Your goal is to recognize the effect software demonstrates and pinpoint the
key functions or operations, necessary to replicate this distinctive pattern."

  PROMPT += """
**High-Level Planning**:
Distill the process into essential stages or components, emphasizing the unique functions
or operations, such as a specific design technique. Concentrate on brevity and precision in
describing each stage, highlighting the unique aspects that contribute to the overall effect.

Please format your response as follows (we use Powerpoint as an example):
'''
1: Insert a Circle and Change its color as black.
2: Add Text 'Happy' inside the Circle.
3: Apply the 'Fly-in' animation for the Circle.
'''

Each stage should be concise yet comprehensive, focusing on the key functionalities or
operations that lead to the visual outcome in PowerPoint. Notably, avoid detailed step-by-step
actions. Strive to keep the number of stages as few as possible, only including those that are
crucial for achieving the unique effect.
"""

  if txt:
      PROMPT += f"**This is the textual descriptions** {txt}"
  return PROMPT
```

**Table 18: High-level Planning Prompt** conditioned on the interleaved instruction query.

**Action – Click.** In Tab. 20, we show the template used by LLM to estimate click coordinates based on image resolution. With SoM's assistance, we use the Tab. 21 template to predict the mark index. With OCR's assistance, we use the Tab. 22 template.

**Action – Drag.** In Tab. 23, we show the template used by LLM to estimate drag coordinates based on image resolution. With SoM's assistance, we use the Tab. 24 template to predict the start and end mark index. With OCR's assistance, we use the Tab. 25 template.

```
def get_prompt(vis=True, txt=None, software=None):

    PROMPT = f"You have been assigned the task of planning a sequence of actions in
{software} software to achieve a desired goal state based on certain conditions.   Your
objective is to outline the fundamental actions needed."

    if vis and not txt:
        PROMPT += "**You are provided with two screenshots which indicate the initial state
as well as goal state.**"

    elif vis and txt:
        PROMPT += "**You are provided with a screenshot to indicate your initial state.**"

    if txt:
        PROMPT += f"**The goal is: {txt}**"

    PROMPT += """
Please format your response as follows:
‘‘‘
1. Click the 'xxx'.
2. Type 'yyy'.
3.: Click the 'zzz'.
’’’
Ensure that each step is clearly described to facilitate step-by-step reproduction of the actions.
"""

    return PROMPT
```

Table 19: **Middle-level Planning Prompt** conditioned on the interleaved instruction query.

I'm working on a computer task that involves clicking on some elements (like a button).
You are provided with a screenshot with a resolution of width: {width} and height: {height}.
Could you assist me in navigating to the "{element}"?
Please provide the location in the following format:
"‘ [x, y] ’"
Ensure that your response contains only the coordinates.

Table 20: **Click action template** that prompts LLMs output click's coordinate [x,y]

The screenshot has been divided into areas and marked with numbers. Where is {element}?
Answer by mark index like [x].

Table 21: **Click action template** that prompts LLMs (with SoM [32]) output coordinate.

**Action – Type / Press.** In Tab. 26, we present the template used by LLM to generate pyautogui code for keyboard actions.

**Action – Scroll.** In Tab. 27, we present the template used by LLM to predict scroll action, which is used for high-level planning. For mid-level planning, we remove the commentary component.

**Evaluation.** In Tab. 28, we display the evaluation template for GPT-4-Turbo [37].

I'm working on a computer task that involves clicking on some elements (like a button). Below are the OCR detection results (element name - bounding coordinates [[x1, y1], [x2, y2]]), which are separated by a colon ";".
{ocr_result}
Could you assist me in navigating to the "{element}"?
Please provide the location in the following format:
"' [x, y] '"
Ensure that your response contains only the coordinates.

**Table 22: Click action template** that prompts LLMs (with OCR [47]) output click's coordinate [x,y]

I am working on a computer task that involves dragging elements from one place to another
You are provided with a screenshot with a resolution of width: {width} and height: {height}.
Could you assist me in navigating for action "{narration}"?
Please provide the location in the following format:
"' [x1, y1] -> [x2, y2] '"
where [x1, y1] are the start coordinates and [x2, y2] are the destination coordinates.
Ensure that your response contains only the coordinates.

**Table 23: Drag action template** that prompts LLMs output drag's coordinate [x1,y1] -> [x2, y2].

The screenshot has been divided into areas and marked with numbers.
To assist with dragging an item, please provide the start and end mark numbers.
How to {element}? Provide the mark indices as follows:
"' [x]->[y] '"
where [x] represents the starting index and [y] represents the ending index.

**Table 24: Drag action template** that prompts LLMs (with SoM [32]) output SoM mark.

I am working on a computer task that involves dragging elements from one place to another
Below are the OCR detection results (element name - bounding coordinates [[x1, y1], [x2, y2]]), which are separated by a colon ";".
{ocr_result}
Could you assist me in navigating for action "narration"?
Please provide the location in the following format:
"' [x1, y1] -> [x2, y2] '"
where [x1, y1] are the start coordinates and [x2, y2] are the destination coordinates. Ensure that your response contains only the coordinates.

**Table 25: Drag action template** that prompts LLMs (with OCR [47]) output drag's coordinate [x1,y1] -> [x2, y2].

I'm working on a computer task involving typing or pressing keys.
Could you assist me in crafting a Python script using pyautogui to accomplish {goal}? where the key input element is "{element}".
I've already set up the environment.
Please provide the executable code directly and refrain from including other outputs or additional code blocks. Ensure that your response contains only one code block formatted as follows:
'''python
import pyautogui
pyautogui.press('ctrl')
'''

**Table 26: Type / Press action template** that prompts LLMs output `pyautogui` code.

I'm currently engaged in a computer-based task and need your assistance.
You are provided with an image of my screenshot.
Could you advise whether I need to scroll to see the complete element "{element}"? Please note that even if the element appears partially, I still need to scroll to see it completely.

'A': 'No need to scroll.', 'B': 'Scroll down.', 'C': 'Scroll up.'

Please select the appropriate option and format your response as follows (Wrap options in square brackets):
''' [A] '''

**Notably, only output options with square brackets**

**Table 27: Scroll action template** that prompts LLMs to output a decision like scrolling (up/down) or not.

You are tasked with evaluating the quality of a software procedure plan. Assess the prediction provided by an AI model against the human-generated ground truth and assign a correctness score to the prediction.

**Evaluation Criteria:**

1. *Conciseness and Clarity*: The procedure plan should be straightforward and to the point.

2. *Element Accuracy*: Pay attention to the precision of specific details like types of animation, text content, and design elements (e.g., 3d shape, color, shape). The prediction should accurately reflect these aspects as mentioned in the ground truth.

3. *Commentary*: Provide a brief commentary in your response summarizing the accurate and inaccurate aspects of the prediction as evidence to support your scoring decision.

**Correctness Score** (must be an integer):

- 0: Completely incorrect

- 1 to 3: Partially correct (with 1 being least accurate and 3 being more accurate)

- 4 to 5: Fully correct (with 4 being good and 5 being perfect)

**Ground truth:**
{GT}

**Prediction:**
{Pred}

Considering the detailed elements and the overall process, please format your response as follows:
```
[comment]:  Summary of evaluation.
[score]:  x
```

**Table 28:** Evaluation Prompt Template

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
