# OpenReview forum: "VideoGUI: A Benchmark for GUI Automation from Instructional Videos"
_NeurIPS.cc/2024/Datasets_and_Benchmarks_Track — NeurIPS 2024 Track Datasets and Benchmarks Spotlight_

### Official Review · Reviewer_ymcv · 2024-07-05
**VideoGUI**

**Rating:** 6
**Confidence:** 3
**Correctness:** See my comments on 'Opportunities For…
**Clarity:** Yes.

**Review:**

The authors propose a benchmark of 86 complex tasks in 11 visual-centric software applications. The tasks are extracted from instructional videos.

The idea of using instructional videos as a source for creating a benchmark for GUI automation is interesting and promising. The authors split up the benchmark into three levels: high-level planning (visual query), mid-level planning (visual+text or text-only) and atomic actions.

Pros
- I think the collected data and the proposed hierarchical evaluation are useful.

Cons
- It is unclear if/how the benchmark can handle alternative ways of executing a task.
- It is unclear how the high and mid-level planning are evaluated.

**Strengths:**

I think the collected data and the proposed hierarchical evaluation are useful and novel. They can be helpful for GUI automation researchers to evaluate their approaches.

**Additional Feedback:**

Q1) 'Visual query is the primary setting'  - How does this work for software like Premiere Pro which has a video as output?
Q2) What about alternative ways to execute a task? (e.g. using a keyboard shortcut)
Q3) How sensitive are the dist metrics, in particular the one for Drag? There is usually a large area that responds to this action.

**Documentation:**

There are supplementary files but I could not find the actual dataset or information about how it will be distributed.

**Ethics:**

No.

**Limitations:**

As far as I could tell, the limitations of the benchmark are not described in the paper (only in the checklist).

**Opportunities For Improvement:**

- The scores of current LMMs are low, especially for high-level plans. But in many software applications, there are several ways to achieve an outcome (e.g., through keyboard shortcuts, or menu vs. toolbar access). In many cases (as far as I am aware), instructional videos show only one way to achieve an outcome. How does the benchmark account for such alternatives? And more importantly, if an alternative is unknown by the benchmark, how does it affect the benchmark score?

- It is unclear how the metrics used to evaluate high and mid-level planning are computed. Lines 150-154 state that they are evaluated on a scale from 0 to 5, but e.g. Table 3 also evaluates them using a %. What does this % indicate? Also - how is the scale from 0 to 5 defined?

**Relation To Prior Work:**

Yes.

**Summary And Contributions:**

The authors propose a benchmark of 86 complex tasks in 11 visual-centric software applications. The tasks are extracted from instructional videos.

---

> ### Author Rebuttal · Authors · 2024-08-17
>
> > We appreciate that you found our method interesting and promising. We notice your major concern on alternative ways for a task. We value this opportunity to clarify and address your concerns.
> # C1&O1&Q2: High-level plan scores are low. As instructional videos show only one way to achieve an outcome, How does benchmark account for an alternative way?
> We agree with the reviewer’s point that there might be multiple action trajectories to accomplish one task shown in instructional videos. However, it is important to note that **these videos, typically created by experts, often demonstrate the most common and representative approach to completing a task.**
>
> Moreover, we have highlighted this concern (main text line 150-151) and **provide several strategies:**
> 1. Firstly, we need to clarify that **the goal of high-level planning (line 142) is for recognizing the key milestone** (e.g., particular animation type) rather than detailed action sequence. So `low high-level plan scores are mainly due to models failing to recognize rather than alternative pathways`. The alternative issue is mainly focused on **middle-level planning**.
> 2. **To address this at the planning stage**, We incorporate the **LLM Critic** (lines 150-151) to account for `human-like subjective reasoning`. This method is capable of considering alternative actions, such as recognizing that `Ctrl + C` is equivalent to `Right-click + Copy`.
> 3. **To address this at the action execution stage**, we strictly limiting the action and associated contexts, providing precise instructions such as `RightClick the Red Rectangle,` accompanied by a matched screenshot.
> 4. **To allow any alternative planning possibilities as long as the model successfully achieves the final outcome**, we include the metric of Success Rate, as demonstrated in Supp. Tab. 13-14. This evaluation approach inherently supports alternative planning trajectories.
> 5. **A possible future effort** is that we `provide diverse enough planning annotations by multiple annotators for the same task`.
>
> # C2: How is high and mid-level planning evaluated?
> We use **LLMs as a Critic** by `GPT-4-Turbo`, which is inspired by the recent works in multi-modal LMM benchmarks – MM-Vet (ICML'2024). The LLM Critic accepts inputs:
> 1. The ground-truth planning by human.
> 2. The predicted planning by models.
> 3. A carefully designed system prompt, which guides LLM as a evaluator (which we clarify in Supp. 1.4)
>
> Then output a score (0–5).
>
> # O2: How to convert the planning scores (0-5) in Table 3 into %? How is the scale from 0 to 5 defined?
> We apologize for any confusion caused. The planning score was originally evaluated on a scale from 0 to 5. However, in Table 3, we aimed to **provide an overall score that considers both planning and action execution**, with the latter being evaluated on a scale of 0 to 1. To achieve this, we **normalized** the planning scores by dividing by the maximum value of 5.
> We will include additional clarification in the revised version.
>
> For detailed definitions of the 0 to 5 score range, please **refer to our Supp. section 1.3, Table 11**, where we explain the meaning of each score (We attach it below).
>
> ```
> Correctness Score (must be an integer):
> - 0: Completely incorrect
> - 1 to 3: Partially correct (with 1 being least accurate and 3 being more accurate)
> - 4 to 5: Fully correct (with 4 being good and 5 being perfect)
> ```
>
> # Document: Are the datasets and codes available?
> So far, we **have uploaded the following materials**:
> 1. **Dataset**:
>     - **High-level Planning:** https://huggingface.co/datasets/VideoGUI/VideoGUI-High-Plan
>     - **Middle-level Planning:** https://huggingface.co/datasets/VideoGUI/VideoGUI-Mid-Plan
>     - **Action Execution:** https://huggingface.co/datasets/VideoGUI/VideoGUI-Action
> 2. **Codes**: https://anonymous.4open.science/r/VideoGUI-C05D/README.md
> 3. **Datasheet Document**: https://drive.google.com/file/d/1cIoQCX_YBsWBfuCF08vUE1foSvQpFgfn/view?usp=sharing
> 4. **Website (Human vs. Agent)**: https://videogui.github.io/
>
> # Q1: How does Visual query work for software (e.g., Premiere Pro) which has a video as output?
> In this scenario, we handle the video output from software like Premiere Pro by providing a video preview (e.g., multiple frames) as the input, paired with a task-specific query such as: `[video frames] How to create this effect in {Premiere Pro}?`
> The **prompt template for this setup is detailed in Supp. Section 1.3, Table 1**. In this context, only the `vis_end` is provided as input, without `vis_start` or `text` terms in the function input.
>
> # Q3: How sensitive are distance metrics for drag? There is usually a large area that responds to this action.
>
> - **Firstly, to address the concern about `large area`**. It is important to clarify its meaning. If the 'large area' indicates to the distance between the starting and ending points of a drag action, our metric remains unaffected. This is because the metric **independently considers proximity to the start and end points.** Our method employs precise action narration, which ensures that operations at both the start and end points are specifically defined thus reducing the area space. For example, `Drag [the top-left corner of the Circle]`. **Further details on this strategy can be found in our response to Reviewer Pf3N-O3.**
>
> - **Secondly, regarding the sensitivity of drag metrics**, we acknowledge the inherent challenges and have developed two types of metrics: `Absolute distance – a continuous metric`, and `Recall @ d – a discrete metric`, where the selection of `d` is empirically estimated across many softwares.
> Our findings indicate that the model (i.e., GPT-4o) achieves the smallest distance and the highest Recall, demonstrating **a general consistent trend across these metrics**, thus serving as an effective signal**.
>
> |Model|**Dist↓**|Rank by Dist|**Recall↑**|Rank by Recall|
> |-|-|-|-|-|
> |**CogAgent**|44.7|3|0|3|
> |**GPT-4-Turbo**|31.3|2|1.7|2|
> |**GPT-4o**| 21.9|1|2.5|1|

---

> > ### Comment · Reviewer_ymcv · 2024-08-25
> >
> > Thanks for the response!
> >
> > I still think that the benchmark measures the capability of a model to execute a task in a certain way (instead of executing a task, like the authors claim). But it seems that the authors have taken measures to at least prevent that alternative executions do not negatively affect the benchmark score. I encourage the authors to discuss these measures more clearly in the revised paper.
> >
> > With these measures in place, I think the paper makes a nice contribution. So I'll increase my score.

---

> > > ### Author Response · Authors · 2024-08-26
> > > **Thanks for acknowledging our contributions!**
> > >
> > > Dear Reviewer,
> > >
> > > Thank you for your valuable feedback and for acknowledging our contributions. We greatly appreciate your suggestions and feel encouraged by your response.
> > >
> > > In our benchmark, *recognizing the challenges posed by complex full tasks, we pursued an effective approach that provides rich feedback for models*. This led us to develop a hierarchical strategy with three stages, where each stage assesses the model's ability to execute a task at different levels—ranging from planning to action, which includes considerations like (i) use LLM critic for planning (ii) detailed action narration with matched screenshot (iii) include Success Rate evaluation. Potential improvement can be (iv) diverse annotations per task by multiple annotators.
> > >
> > > We promise to include a more clear discussion regarding these measures in our revised paper. Thank you!

---

### Official Review · Reviewer_YcXa · 2024-07-22

**Rating:** 6
**Confidence:** 5
**Correctness:** Yes.
**Clarity:** Yes.

**Review:**

Overall this is a quite good paper, which considers the fact that humans frequently rely on instructional videos to master complex skills. It also considers 3 different query types for the first time, which is quite reasonable. The writing, illustration, and experiment design of the paper is also very good.



Pros:

- The paper takes into account the fact that humans often rely on instructional videos to master complex skills and derive tasks from these videos.
- It considers 3 different query types for the first time, which is quite reasonable.
- The writing and illustration of the paper is good.
  - The comparsion with previous works is clear.
  - The illustration of the statistics of the benchmark.
- The experiment design and the systematical evaluation over different models is solid.
- A novel hierarchical evaluation process is proposed to better identify detailed failure causes.

Cons:

- What is the environment in which these tasks are evaluated and performed? Do users need to use their own computers as the environment?
- What's the principle for selecting these teaching videos from so many videos?
- I haven't found any link or evidence that this benchmark is or will be open source, since the authors claim that "We will provide code, data samples and instructions in supplementary material.".

**Strengths:**

The paper takes into account the fact that humans often rely on instructional videos to master complex skills and derive tasks from these videos; It considers 3 different query types for the first time, which is quite reasonable; The writing and illustration of the paper is good; The experiment design and the systematical evaluation over different models is solid; A novel hierarchical evaluation process is proposed to better identify detailed failure causes.

**Additional Feedback:**

N/A

**Documentation:**

I haven't found any link or evidence that this benchmark is or will be open source, since the authors claim that "We will provide code, data samples and instructions in supplementary material.".

**Ethics:**

No.

**Limitations:**

Yes.

**Opportunities For Improvement:**

What is the environment in which these tasks are evaluated and performed? Do users need to use their own computers as the environment? What's the principle for selecting these teaching videos from so many videos?

**Relation To Prior Work:**

Yes.

**Summary And Contributions:**

The paper proposes VideoGUI, a GUI agent benchmark to evaluate LLM/MLLM agents on complex and visually-intensive software tasks. The benchmark includes 86 complex tasks and 463 subtasks, derived from high-quality web instructional videos. The most important feature is that VideoGUI considers 3 different query types for the first time, namely text, image and video. A novel hierarchical evaluation process is also proposed in the benchmark, which includes high-level planning, middle-level planning, and atomic action execution. The authors systematically evaluate 9 different models on the benchmark. Some takeaways of the experiment are summarized.

---

> ### Author Rebuttal · Authors · 2024-08-17
>
> > Thank you for your positive feedback and for recognizing the value of our proposed method! Below we address your concerns:
> # C1: What’s the environment for evaluations? Do users need to use their own PC as the environment?
> | Eval. style                                   | Environment Platform | Metrics                        | Reference |
> |-----------------------------------------------|----------------------|--------------------------------|-------------------|
> | [Primary] Human annotations + LLM as CRITIC             | No required (e.g., Linux)         | Three stages individual scores | Main body, Table 3 |
> | Agent execution + Human success verify        | Real Simulator (Window)       | Success Rates                  | Supp. Tables 13 and 14 |
>
>
> - As shown in the above table, we employed two evaluation approaches in our paper. **The primary setting (Main body, Table 3)** divides the evaluation into three distinct stages, with inputs and annotations prepared as ground truth. This allows both planning and action execution evaluations to be treated as `offline question-answering tasks, conditioned by the visual inputs or context`. Therefore, this method **does not require** the same environment. We conducted these evaluations in a `Linux terminal`.
> - However, if users wish to replicate the **Success Rate settings described in Supp. Tables 13 and 14**, they **will need** the same environment with the required software installed. We run agent in the `Windows desktop` environment.
>
> # C2: Principle for selecting instruction videos
> We selected the videos based on both topic relevance and quality:
>
> **By Topic:**
> - Videos introducing novel concepts or features with visual preview effects, primarily for visual creation and editing software.
> - Videos offering advanced knowledge beyond basic usage, such as "Top tips" for VLC Player.
>
>
> **By Quality:**
> - High-resolution videos with clear, step-by-step instructions.
> - High-quality, accessible transcripts that users can easily follow.
>
> We will include them in our revision.
>
> # C3: Are the datasets and codes available?
> So far, we **have uploaded the following materials** for reviewers to access:
> 1. **Dataset (Huggingface for easy access)**:
>     - **High-level Planning:** https://huggingface.co/datasets/VideoGUI/VideoGUI-High-Plan
>     - **Middle-level Planning:** https://huggingface.co/datasets/VideoGUI/VideoGUI-Mid-Plan
>     - **Action Execution:** https://huggingface.co/datasets/VideoGUI/VideoGUI-Action
> 2. **Project Codes**: https://anonymous.4open.science/r/VideoGUI-C05D/README.md
> 3. **Datasheet Document (data document, intended use, maintainence plan, license, author statement, etc)**: https://drive.google.com/file/d/1cIoQCX_YBsWBfuCF08vUE1foSvQpFgfn/view?usp=sharing
> 4. **Website (Human vs. Agent)**: https://videogui.github.io/

---

> > ### Author Response · Authors · 2024-08-27
> > **Gentle Reminder to Review Our Rebuttal**
> >
> > Dear Reviewer YcXa,
> >
> > Thank you once again for your feedback!
> >
> > Since the rebuttal period is halfway through, we would greatly appreciate it if you could review our response to ensure it adequately addresses your concerns.
> >
> > Thank you for your time and consideration.
> >
> > Best regards,
> >
> > Authors of 693

---

### Official Review · Reviewer_pM4s · 2024-07-25
**Good paper**

**Rating:** 8
**Confidence:** 4
**Clarity:** The paper is well-written and easy-to…

**Review:**

Generally, this paper provides a novel scenario for GUI automation, which is reasonable and useful in real-world scenarios. Learning from instructional videos is an essential ability for both humans and machines. To achieve this goal, the authors designed a thorough data curation pipeline and proposed detailed evaluation metrics at three different levels, providing a strong platform for GUI automation analysis. Please see the detailed pros and cons in the following sections.

**Strengths:**

1. Motivation: The authors provide a novel scenario for GUI automation, i.e., learning from instructional videos, which is reasonable and useful in real-world scenarios.
2. Annotation: The data generation process went through a clear and rigorous pipeline that includes manual annotation and quality checks.
3. Evaluation: To better evaluate existing methods, the authors also proposed a hierarchical evaluation process that considers at which level a model might fail. This can largely help analyzing these methods and inspire future improvements.

**Additional Feedback:**

I think this is a good paper on GUI automation but have a small concern regarding the evaluation process. I would also like to hear more discussions on the difficulties of developing a fully automatic evaluation pipeline, and whether there are any possible solutions, which would make this submission much stronger and self-contained.

**Correctness:**

The proposed hierarchical evaluation requires human and LLM assistance, which might bring some uncertainty that leads to unreliable results.

**Documentation:**

The authors provided sufficient detail about the dataset.

**Limitations:**

The authors have clearly discussed the limiations and potential negative societal impact of their work in the checklist.

**Opportunities For Improvement:**

1. It would be better if the authors could include the evaluation results of Gemini-1.5-Pro, which has been proven to have excellent video understanding abilities.
2. The evaluation process is not fully automatic, which requires human and LLM assistance. It would be better if the authors could provide some insights about providing a fully automatic while extensive evaluation.

**Relation To Prior Work:**

This work significantly differs from existing ones in terms of both task formulation and evaluation method.

**Summary And Contributions:**

This paper introduces VideoGUI, a comprehensive multi-modal benchmark for GUI automation. The proposed benchmark significantly differs from existing ones in terms of two aspects: 1) [Task instruction] The proposed benchmark leverages long instructional videos rather than short and simple text queries as conditions for GUI automation. 2) [Evaluation] Authors proposed a novel hierarchical process to do the evaluation at three levels, i.e., high-level planning, middle-level planning, and atomic action execution. Based on this dataset, the authors conducted an extensive evaluation of existing LMMs and revealed the gap in visual conditioned planning that shows the direction for developing stronger GUI automation systems.

---

> ### Author Rebuttal · Authors · 2024-08-17
>
> > We are very encouraged by your positive response! Your comments on Human and LLM evaluations are very inspiring. Below we have addressed your questions:
> # O1: Would be better to include Gemini-1.5-Pro
> Thanks! Gemini-1.5-Pro is a worth-trying model and we have tested it, compared with Gemini-Pro, it has made significant gains in each dimension, with an overall 4.6 improvement.
>
> |                    | High-level Planning | Mid-level Planning | Action Exec. | **Overall**      |
> |--------------------|---------------------|--------------------|--------------|------------------|
> | **Gemini-Pro-V**   | 7.9                 | 28.6               | 23.8         | 27.7             |
> | **Gemini-1.5-Pro** | 11.4                | 46.8               | 38.7         | 32.3 (**+4.6**)  |
>
> # O2: Evaluation requires Human and LLM assistance, thus is not fully automatic. Would be better to provide some insights about fully automatic.
> 1. In evaluation, accuracy is the most important. As shown in the below Table, while **Human-only** annotation ensures high-quality results, it is extremely time-consuming. On the other hand, **LLM-only** allows for full automation but may lead to issues such as hallucinations, rendering the output potentially unreliable. Consequently, a hybrid **Human+LLM** combining human expertise and LLMs is a reasonable compromise, offering a balanced solution.
>
> |                           | **Pros**                                                       | **Cons**                                                       | **Environment Platform**   |
> |---------------------------|---------------------------------------------------------------|----------------------------------------------------------------|----------------------------|
> | **Human-only**            | High-quality, Interpretability                                | Extremely cost                                                 | Real Simulator             |
> | **LLM-only**              | Fully automatic                                                | Hallucinations, might be unreliable                            | No required.               |
> | **Human & LLM** *(Human annotations + LLM as CRITIC)*           | Comprehensive signals for each stage; Automatic once we collected all annotations | Require annotations for each task in advance.                   | No required.               |
> | **Human & LLM** *(Agent execution + Human success verify)*           | Check whether agent indeed complete the full-task              | Require human check output (but is fast)                       | Real Simulator             |
>
> 2. Besides, in Table **row 3** i.e., `·after obtaining annotations for each task, our main evaluation pipeline actually doesn't need human assistance`, and it is **an acceptable automatic solution**.
> 3. Nevertheless, there are opportunities for further refinement. For instance, we could **develop task-specific pipelines tailored to individual outcomes.** This could involve feeding the final output generated by the agent into a verification process, which applies specific rules to assess whether the expected content (e.g., text, shapes) is present at the desired location. These potential enhancements will be considered in future work. :)
>
> # Correctness: Hierarchical evaluation requires human and LLM assistance, might bring uncertainty.
> - We recognize that incorporating LLMs in the evaluation process may introduce some degree of uncertainty. Currently, we prompt LLMs output by setting the temperature to zero and fixing the API version and prompt templates.
> - While we do not overemphasize on the scores value meaning generated by the LLM Critic, these scores **do provide a useful indication of trends and allow us to compare the relative strengths of models A and B.**
> - To further demonstrate the reliability of the LLM Critic scores, on High-Level Planning, we assign a CS graduate student and include the manual scores for comparison in the below Table.
>
> | High-Level Planning | CRITIC (0–5) | Human (0–5) |
> |---------------------|--------------|-------------|
> | **GPT-4o**          | 0.86         | 1.24        |
> | **Claude-3-Opus**   | 0.48         | 0.62        |
> | **Gemini-Pro**      | 0.39         | 0.46        |
>
> Despite their differences, **both LLM Critic and Human scores maintain the same trend (e.g., ranking among models).**
> Beside, we interestly found that human tend to assign higher score than LLM critic. This might be due to the model performing poorly in high-level planning, while `the same person, after repeatedly scoring, tended to accept related but not entirely consistent answers to account for variation` (e.g., considering related animation types as correct). In contrast, `the LLM critic scores each attempt independently, without historical memory, making it much stricter`. This also reflects **the bias inherent in human evaluation.**

---

> > ### Comment · Reviewer_pM4s · 2024-08-22
> >
> > Thanks for the detailed response. The authors' rebuttal has addressed all my concerns. I'm keeping my original rating as 8 (clear accept).

---

> > > ### Author Response · Authors · 2024-08-22
> > > **Thanks for your positive feedback**
> > >
> > > Dear Reviewer,
> > >
> > > We are glad that our response has addressed all your concerns.
> > >
> > > Thank you for your constructive feedback and for recognizing the value of our work!

---

### Official Review · Reviewer_Pf3N · 2024-07-26
**New Benchmark for GUI Automation from Instructional Videos**

**Rating:** 6
**Confidence:** 3
**Clarity:** The paper is clear, well written and …

**Review:**

Overall, this paper is clear, well-motivated and provides a new benchmark for GUI automation based on instructional videos. The tasks selected from the benchmark are mostly in the creative space, which involve professional, novel software and complex activities, contributing to the diversity and novelty of the agent benchmark tasks. In addition, analogous to human learning from instruction videos, understanding how agents learn directly from videos or visual instructions (in particular in creative tasks) is important to agent development. The paper is also able to break down the tasks into both planning (high level and low level), and execution phase, which gives more clarity on agent performance.

It is interesting to note that current multi-modal models struggle with GUI tasks, specifically generating coordinates to manipulate the GUI interface - which is more obvious in the dragging operation. Planning seems to be another bottleneck for agent performance, but a lot also due to the fact that many of the key instructions in the planning phase include esoteric terms such as "morph transition". It would be interesting to compare agents with human performance in those tasks without training videos.

One major weakness is that the authors have not provided code and detailed list of tasks for reproducibility, which is crucial for benchmarks.

**Strengths:**

One of the contributions of the paper is that it provides a new benchmark for complex, realistic creative processes that involve novel professional tools. Especially for tasks such as media editing, instructional video is complex and it is non trivial to describe the visual requirements with text. Also, it is worth noting that drag as an action is a lot more common for media editing / creative tasks than most of other professional tasks, which adds more diversity to the existing benchmarks.

In addition, analogous to human learning from instruction videos, understanding how agents learn directly from videos or visual instructions (in particular in creative tasks) is important to agent development. The paper is also able to break down the tasks into both planning (high level and low level), and execution phase, which gives more clarity on agent performance.

The authors are also able to provide video summary and documentation of video recording between human and agent executing the sample task, which is intuitive for readers.

**Additional Feedback:**

Missing a bracket for line 33 - the distance formula in the supplementary material.

**Correctness:**

To the best of my knowledge, there are no correctness issue regarding benchmark evaluation methods and experiment design. I have also voiced my other concerns in the sections above.

**Documentation:**

The authors provided details, but not yet code and GitHub repository to support reproducibility.

**Ethics:**

Since the authors collected instructional videos with transcripts from YouTube, there might be copyright concerns from the original video creators. There are no further ethical concerns regarding this paper to the best of my knowledge.

**Limitations:**

The authors discussed briefly that due to the high collection cost, VideoGUI is limited by its smaller scale. The authors could provide a more detailed discussion in limitations such as the evaluations are mostly on a single platform (desktop),  lack of guardrails, etc.

**Opportunities For Improvement:**

The paper could benefit from discussions with regards to the following points:
1. For documentation, there is no project code and full list of tasks published for reproducibility. Could the authors provide relevant details?
2. Agent critic for planning task evaluation always brings the question of bias and lack of interpretability. Would it be better if there is also human evaluation as reference?
3. It is not clear to readers sometimes on the definition of ground truth coordinates - especially for the task of dragging. How do you define the starting and end points, especially that none of the ground truth plans define the exact distance to be dragged?
4. Even though models still have a lot of space for completing the tasks, are there considerations around robustness to see if the models could perform the tasks consistently?
5. It is interesting to note that current multi-modal models struggle with GUI tasks, specifically generating coordinates to manipulate the GUI interface - which is more obvious in the dragging operation. Planning seems to be another bottleneck for agent performance, but a lot also due to the fact that many of the key instructions in the planning phase include esoteric terms such as "morph transition". It would be interesting to compare human performance in those tasks without training videos.

**Relation To Prior Work:**

To the best of my knowledge, this work discusses how it differs from previous contributions.

**Summary And Contributions:**

This paper introduces VideoGUI, a multi-modal benchmark designed to evaluate GUI assistants learning from instructional videos. VideoGUI focuses on tasks involving professional, novel software and complex activities in the creative space, through a hierarchical process of high-level, middle-level planning, and action execution. The benchmark shows performance gaps in planning from visual previews and certain actions such as dragging.

---

> ### Author Rebuttal · Authors · 2024-08-17
>
> > Thank you for recognizing the value of our work and for providing constructive and thorough feedback. We greatly value your insights and hope our response addresses your comments.
> # O1: Are the datasets and codes available?
> So far, we **have uploaded the following materials**:
> 1. **Dataset**:
>     - **High-level Planning:** https://huggingface.co/datasets/VideoGUI/VideoGUI-High-Plan
>     - **Middle-level Planning:** https://huggingface.co/datasets/VideoGUI/VideoGUI-Mid-Plan
>     - **Action Execution:** https://huggingface.co/datasets/VideoGUI/VideoGUI-Action
> 2. **Codes**: https://anonymous.4open.science/r/VideoGUI-C05D/README.md
> 3. **Datasheet Document**: https://drive.google.com/file/d/1cIoQCX_YBsWBfuCF08vUE1foSvQpFgfn/view?usp=sharing
> 4. **Website (Human vs. Agent)**: https://videogui.github.io/
>
> # O2: Human evaluation as planning tasks reference might overcome bias and lack of interpretability.
> Thank you for your insightful suggestion!
>
> **(i)** We have indeed considered interpretability **in our LLM Critic**. Specifically, we require it to generate **Commentary** that supports its score decisions, as detailed in Supp. Sec. 1.4, Table 11.
>
> **(ii)** Moreover, we conducted **manual verification (Success Rate)** outlined in Supp. Section 3, where human evaluators verify the final output and calculate success rates. **Focusing on the final output is more direct and concise** instead of evaluating the planning process for human.
>
> **(iii)** We agree with the reviewer that human evaluation can be valuable. But it being time-intensive to evaluate each model in every stage. Here, we assign a graduate student to score high-level planning follow the same scoring criteria (Supp. Sec. 1.3, Tab. 1). And We found despite have difference, the **LLM Critic maintains the same trend (ranking) as Human**. This suggests that LLM Critic has the potential to become an auto. eval tool
>
> |High-Level Planning|Critic (0–5)|Human (0–5)|
> |-|-|-|
> |**GPT-4o**| 0.86|1.24|
> |**Claude-3-Opus**|0.48|0.62|
> |**Gemini-Pro**|0.39|0.46|
>
> # O3: How to define starting and end points for dragging? Especially that none of the ground truth plans define the exact distance to be dragged?
>
> Good question!
> As there are no GT plans for drag actions in the instruction videos, we address this by **recording human drag demonstrations** and **capture drag actions by precise narrations.**
>
> Notably, we ask annotators to provide a textual quadruple for each drag action: ``[start position, end position, element, purpose]``. The narration follows the format: `Drag the [element] from [start position] to [end position] to [purpose].` Here, the `start` and `end positions` guide the annotators in identifying locations within the screenshot, with the `element` representing the object being dragged and the purpose defining the `goal` of the action (mainly movement or resizing)
>
> - For **movement**, the start point is usually the element’s original parent element (or "original position" if unspecified), while the end point is determined by the target parent element (e.g., a panel).
> - For **resizing**, the start point is based on the specific part of the element being dragged (e.g., “top-left corner of the circle”), and the end point is identified by relevant elements in the screenshot.
>
>
> Additionally, our focus is on **making predictions that closely approximate the intended points, rather than matching exact coordinates**, which is why the narration serves as an effective guide.
>
> # O4: Considerations around robustness for model consistency?
> Currently, we generate LLM outputs using fixed prompts, a temperature of zero, and a fixed API version to minimize uncertainty. We agree with the reviewers that robustness is important.
> Therefore, we run high-level planning multiple times and calculate their average scores along with variance.
>
> |High-Level Planning (0–5)| Single Run | Multiple Run (3 times) |
> |-|-|-|
> |**GPT-4o**|0.86| 0.89 (±0.11)|
> |**Claude-3-Opus**|0.48| 0.52 (±0.15)|
> |**Gemini-Pro**|0.39| 0.37 (±0.14)|
>
> The differences between single and 3 times falling within an acceptable range of error, indicating consistent trend. We will include multiple run experiments in our revision.
>
> # O5: Esoteric terms (e.g., morph transition) might be a bottleneck for agent performance. Compare human performance in those tasks without instruction videos?
>
> Thank you for the insightful question. We first investigated **whether low performance is due to esoteric terms**. **As shown in Fig. 6 (top half)**, LLMs can generate terms like `fly-in` and `"rotate animation`, indicating they own relevant domain knowledge but they struggle to recognize them solely on visual queries.
>
> On the commonly used Powerpoint software, we evaluated human performance by involving a student (without watching videos, general-level) on the `High-level Planning -- Challenging Visual-centric Setting`.
> We found the student's scores were lower, as **student is non-expert**. This should serve as a **lower bound** for agent performance, **as the agent is intended to assist regular users**.
>
> | Variant| High-level planning |
> |-|-|
> | Human (normal people) — Lower bound|1.41|
> | Agent -- GPT-4o| 1.82|
> | Human (software expert) – High bound| N/A|
>
> # L1: Provide more discussion on limitations?
>
> Thanks! we will include more in revision, such as:
> - So far we focus on Windows, but software versions among different platforms might bring differences.
> - Extend to cross-software evaluation. Such as first collecting video assets on a website then editing them in PR / AE.
>
> # Ethics: instructional videos might have copyright concerns from the video creators.
> 1. While we collect instructional videos from YouTube, we **do not use these Youtube videos in our evaluation**. Instead, we provide our human demonstration to models.
> 2. If future work requires videos, we can provide the URLs and direct users to the original platforms.
>
> # Additional Feedback: Missing a bracket in line 33
> Thanks, we have corrected it :)

---

> > ### Comment · Reviewer_Pf3N · 2024-08-28
> >
> > Thanks for the detailed response and I have increased my rating accordingly to reflect the improvements.

---

> > > ### Author Response · Authors · 2024-08-28
> > >
> > > Dear Reviewer Pf3N,
> > >
> > > We are glad to receive your positive feedback. We will include these discussions in the revision.
> > >
> > > Thank you again for your valuable suggestions and time!

---

> ### Author Response · Authors · 2024-08-27
> **Gentle Reminder to Review Our Rebuttal**
>
> Dear Reviewer Pf3N,
>
> Thank you once again for your feedback!
>
> Since the rebuttal period is halfway through, we would greatly appreciate it if you could review our response to ensure it adequately addresses your concerns.
>
> Thank you for your time and consideration.
>
> Best regards,
>
> Authors of 693

---

### Author Rebuttal · Authors · 2024-08-17

# General Summary
We sincerely thank all the reviewers for their time and constructive feedback.
We are encouraged by the reviewers' recognition that:
- **Our motivation is significant and novel** – `Learning from instruction videos is important either for humans or agents` (Pf3N, pM4s, YcXa, ymcv).
- **Our collected annotations are thorough**, with good design of `hierarchy structure`. (Pf3N, pM4s, ymcv)
- **Our experiments are systematic and comprehensive**, with proposed metrics for each aspect, and can reflect which level a model might fail. (Pf3N, pM4s, YcXa, ymcv)
- **Our paper is well-written and easy to follow** (Pf3N, pM4s, YcXa).

# Datasets and Codes
So far, we **have uploaded the following materials** for reviewers to access:
1. **Dataset (Huggingface for easy access)**:
    - **High-level Planning:** https://huggingface.co/datasets/VideoGUI/VideoGUI-High-Plan
    - **Middle-level Planning:** https://huggingface.co/datasets/VideoGUI/VideoGUI-Mid-Plan
    - **Action Execution:** https://huggingface.co/datasets/VideoGUI/VideoGUI-Action
2. **Project Codes**: https://anonymous.4open.science/r/VideoGUI-C05D/README.md
3. **Datasheet Document (data document, intended use, maintainence plan, license, author statement, etc)**: https://drive.google.com/file/d/1cIoQCX_YBsWBfuCF08vUE1foSvQpFgfn/view?usp=sharing
4. **Website (Human vs. Agent)**: https://videogui.github.io/

Please find our detailed responses to your specific questions below. For clarity, we use the following notations:
- O – Opportunities For Improvement;
- L – Limitations
- C - Cons
- Q - Question

---

### Decision · Program_Chairs · 2024-09-26

**Decision:**

Accept (Spotlight)

**Comment:**

All four reviewers recommend acceptance, with one being particularly enthusiastic about this paper. The Area Chair agrees, and finds the topic to be especially creative. Congratulations to the authors.